# StaQ: a Finite Memory Approach to Discrete Action Policy Mirror Descent

## Abstract

In Reinforcement Learning (RL), regularization with a Kullback-Leibler divergence that penalizes large deviations between successive policies has emerged as a popular tool both in theory and practice. This family of algorithms, often referred to as Policy Mirror Descent (PMD), has the property of averaging out policy evaluation errors which are bound to occur when using function approximators. However, exact PMD has remained a mostly theoretical framework, as its closed-form solution involves the sum of all past Q-functions which is generally intractable. A common practical approximation of PMD is to follow the natural policy gradient, but this potentially introduces errors in the policy update. In this paper, we propose and analyze PMD-like algorithms for discrete action spaces that only keep the last $M$ Q-functions in memory. We show theoretically that for a finite and large enough $M$, an RL algorithm can be derived that introduces no errors from the policy update, yet keeps the desirable PMD property of averaging out policy evaluation errors. Using an efficient GPU implementation, we then show empirically on several medium-scale RL benchmarks such as MuJoCo and MinAtar that increasing $M$ improves performance up to a certain threshold after which the performance becomes indistinguishable with exact PMD, reinforcing the theoretical findings that using an infinite sum might be unnecessary and that keeping in memory the last $M$ Q-functions is a practical alternative to the natural policy gradient instantiation of PMD.

## 1 Introduction

Deep RL has seen rapid development in the past decade, achieving super-human results on several decision making tasks (Mnih et al., 2015; Silver et al., 2016; Wurman et al., 2022). However, the use of neural networks as function approximators exacerbates many challenges of RL, such as the brittleness to hyperparameters (Henderson, 2018) and the poor alignment between empirical behavior often and theoretical understandings (Ilyas et al., 2020; Kumar et al., 2020; van Hasselt et al., 2018). To address these issues, many deep RL algorithms consider adding regularization terms, one of which being to penalize the Kullback-Leibler divergence (labeled $D_{\mathrm{KL}}$ in the following) between successive policies (Schulman et al., 2015; Wu et al., 2017; Wang et al., 2017; Vieillard et al., 2020b). This family of algorithms is often called Policy Mirror Descent (PMD, Abbasi-Yadkori et al. (2019); Lazic et al. (2021); Zhan et al. (2023)) for its connection—made more explicit in Sec. 4—to the first-order convex optimization method Mirror Descent (Nemirovsky & Yudin, 1983; Beck & Teboulle, 2003). One known property of PMD algorithms, at least in the context of value iteration, it that it averages out policy evaluation errors (Vieillard et al., 2020a). This property can be intuited from the nature of the policy $\pi_k$ at iteration $k$, which is a Boltzmann distribution and its (unnormalized) log-probabilities are a weighted average of past Q-function estimates $\{q_i\}_{i=0}^{k-1}$

$$\pi_k \propto \exp\left(\alpha \sum_{i=0}^{k-1} \beta^i q_{k-i}\right), \tag{1}$$

for temperature $\alpha > 0$ and weight $\beta \in (0, 1]$. In contrast, unregularized RL would pick actions according to $\arg\max_a q_{k-1}(\cdot, a)$, which would be very sensitive to the unavoidable policy evaluation errors introduced by the use of function approximators, that can be in the context of deep RL very large (Ilyas et al., 2020).

While averaging Q-function estimates might cancel out their errors, implementing Eq. 1 exactly might quickly become intractable due to the infinite nature of the sum. This is especially true when a non-linear function approximator is used for $\{q_i\}_{i=0}^{k-1}$, precluding the existance of a compact closed-form expression for the policy. As such, prior work considered several type of approximations to PMD, such as following the natural gradient (Kakade, 2001; Peters & Schaal, 2006; Schulman et al., 2015) or performing a few gradient steps over the regularized policy update loss (Schulman et al., 2017; Tomar et al., 2022). Instead, in this paper we consider a different direction and perform a rigorous study, both theoretically and empirically, of an algorithm—that we call StaQ—that is a weighted average of at most the last $M < \infty$ Q-function estimates, i.e. where the policy is

$$\pi_k \propto \exp\left(\frac{\alpha}{1-\beta^M}\sum_{i=0}^{M-1}\beta^i q_{k-i}\right). \qquad (2)$$

From a theoretical point of view, we show that for $M$ large enough, such a truncation does not introduce a policy update error, i.e. that in the absence of policy evaluation errors, StaQ will converge exactly to the optimal policy. Moreover, StaQ has a similar averaging of error property as PMD, up to some extra terms that decay exponentially fast w.r.t. $M$; suggesting that as we increase $M$, we can quickly recover the behavior of exact PMD. From a practical point of view, the study of a such an algorithm is timely in the age of batched GPU computations: indeed we show that for medium-sized problems such as the continuous state MuJoCo (Todorov et al., 2012) tasks or the image-based MinAtar (Young & Tian, 2019) tasks, increasing $M$ has little to no impact on the run-time, making Eq. 2 a practical alternative policy update to natural policy gradient. This is especially true since the policy update in the discrete action case, to which the scope of this paper is limited to, is optimization free. To summarize, the contributions in this paper are as follow:

1. We extend the PMD analysis of (Vieillard et al., 2020a) in two ways: i) we show that the averaging effect can be obtained by introducing a $D_{\mathrm{KL}}$ penalization only during policy update—instead of during policy evaluation and update, and ii) we extend the analysis to the policy iteration setting, which was analyzed by (Cen et al., 2022; Zhan et al., 2023) without showing the averaging effect of the $D_{\mathrm{KL}}$ regularization.

2. We extend the above analysis to the case of a finite $M$, showing that for $M$ large enough, the policy update is error free, and that the averaging of policy evaluation errors is similar to that of exact PMD up to some extra terms that decay exponentially fast w.r.t. M.

3. We perform an efficient batched implementation of the above algorithm and show that increasing $M$ has beneficial effects on performance, with diminishing returns, to the point that StaQ with a high enough $M$ can become indistinguishable from exact PMD.

## 2 RELATED WORK

**Regularization in RL.** Regularization has seen widespread usage in RL. It was used with (natural) policy gradient (Kakade, 2001; Schulman et al., 2015; Yuan et al., 2022), policy search (Deisenroth et al., 2013), policy iteration (Abbasi-Yadkori et al., 2019; Zhan et al., 2023) and value iteration methods (Fox et al., 2016; Vieillard et al., 2020b). Common choices of regularizers include minimizing the $D_{\mathrm{KL}}$ between the current and previous policy (Azar et al., 2012; Schulman et al., 2015) or encouraging high Shannon entropy (Fox et al., 2016; Haarnoja et al., 2018), but other regularizers exist (Lee et al., 2019; Alfano et al., 2023). We refer the reader to Neu et al. (2017); Geist et al. (2019) for a broader categorization of entropy regularizers and their relation to existing deep RL methods. In this paper, we use both a $D_{\mathrm{KL}}$ penalization w.r.t. the previous policy and a Shannon entropy bonus in a policy iteration context. In Vieillard et al. (2020b), both types of regularizers were used but in a value iteration context. Abbasi-Yadkori et al. (2019); Lazic et al. (2021) are policy iteration methods but only use $D_{\mathrm{KL}}$ penalization.

**Policy Mirror Descent.** Prior works on PMD focus mostly on performing a theoretical analysis of convergence speeds or sample complexity for different choices of regularizers (Li et al., 2022; Johnson et al., 2023; Alfano et al., 2023; Zhan et al., 2023; Lan, 2022; Protopapas & Barakat, 2024). As PMD provides a general framework for many regularized RL algorithms, PMD theoretical results can be naturally extended to many policy gradient algorithms like natural policy gradient (Khodadadian et al., 2021) and TRPO (Schulman et al., 2015) as shown in Neu et al. (2017); Geist et al.

(2019). However, the deep RL algorithms from the PMD family generally perform inexact policy updates, adding a source of error from the theoretical perspective. For example, TRPO and the more recent MDPO (Tomar et al., 2022) rely on approximate policy updates using policy gradients. It was shown in Zhan et al. (2023), that an inexact policy update will add an additional error floor independent of the policy evaluation error. By proposing a finite-memory policy update, we provide new convergence results that offer a new deep RL algorithm policy update step that does not introduce any additional policy update error, in contrast to prior works.

**Ensemble methods and growing neural architectures in RL.** Saving past Q-functions has previously been investigated in the context of policy evaluation. In Tosatto et al. (2017), a first Q-function is learned, then frozen and a new network is added, learning the residual error. Shi et al. (2019) uses past Q-functions to apply Anderson acceleration for a value iteration type of algorithm. Anschel et al. (2017) extend DQN by saving the past 10 Q-functions, and using them to compute lower variance target values. Instead of past $Q$-functions, Chen et al. (2021); Lee et al. (2021); Agarwal et al. (2020); Lan et al. (2020) use an ensemble of independent $Q$-network functions to stabilize $Q$-function learning in DQN type of algorithms. The aforementioned works are orthogonal to ours, as they are concerned with learning one $Q$, which can all be integrated into StaQ. Conversely, both Girgin & Preux (2008) and Della Vecchia et al. (2022) use a special neural architecture called the cascade-correlation network (Fahlman & Lebiere, 1989) to grow neural policies. The former work studies such policies in combination with LSPI (Lagoudakis & Parr, 2003), without entropy regularization. The latter work is closer to ours, using a $D_{\mathrm{KL}}$-regularizer but without a deletion mechanism. As such the policy grows indefinitely, limiting the scaling of the method. Finally, Abbasi-Yadkori et al. (2019) save the past 10 Q-functions to compute the policy in Eq. 1 for the specific case of $\beta = 1$, but do not study the impact of deleting older Q-functions as we do in this paper. Growing neural architectures are more common in the neuroevolution community (Stanley & Miikkulainen, 2002), and have been used for RL, but are beyond the scope of this paper.

**Parallels with Continual Learning.** Continual Learning (CL) moves from the usual i.i.d assumption of supervised learning towards a more general assumption that data distributions change through time (Parisi et al., 2019; Lesort et al., 2020; De Lange et al., 2021; Wang et al., 2024). This problem is closely related to that of incrementally updating the policy $\pi_k$, due to the changing data distributions that each Q-function is trained on, and our approach of using a growing neural architecture to implement a $D_{\mathrm{KL}}$-regularized policy update can be seen as a form of parameter isolation method in the CL literature, which offer some of the best stability-performance trade-offs (see Sec. 6 in De Lange et al. (2021)). Parameter isolation methods were explored in the context of continual RL (Rusu et al., 2016), yet remain understudied in a standard *single-task* RL setting.

## 3 PRELIMINARIES

Let a Markov Decision Problem (MDP) be defined by the tuple $(S, A, R, P, \gamma)$, such that $S$ and $A$ are finite state and action spaces, $R$ is a bounded reward function $R : S \times A \mapsto [-R_{\mathrm{x}}, R_{\mathrm{x}}]$ for some positive constant $R_{\mathrm{x}}$, $P$ defines the (Markovian) transition probabilities of the decision process and $\gamma$ is a discount factor. The algorithms presented in this paper can be extended to more general state spaces. However, we limit ourselves to studying finite action spaces, to simplify the sampling from the Boltzmann distribution for the policy (Eq. 2), which would require deeper algorithmic changes for continuous actions spaces.

Let $\Delta(A)$ be the space of probability distributions over $A$, and $h$ be the negative entropy given by $h : \Delta(A) \mapsto \mathbb{R}$, $h(p) = p \cdot \log p$, where $\cdot$ is the dot product and the $\log$ is applied element-wise to the vector $p$. Let $\pi : S \mapsto \Delta(A)$ be a stationary stochastic policy mapping states to distributions over actions. We denote the entropy regularized V-function for policy $\pi$ and regularization weight $\tau > 0$ as $V_\tau^\pi : S \mapsto \mathbb{R}$, which is defined by

$$V_\tau^\pi(s) = \mathbb{E}_\pi \left[ \sum_{t=0}^\infty \gamma^t \{R(s_t, a_t) - \tau h(\pi(s_t))\} \middle| s_0 = s \right]. \tag{3}$$

In turn, the entropy regularized Q-function is given by $Q_\tau^\pi(s, a) = R(s, a) + \gamma \mathbb{E}_{s'} [V_\tau^\pi(s')]$. The V-function can be written as the expectation of the Q-function plus the current state entropy, i.e. $V_\tau^\pi(s) = \mathbb{E}_a [Q_\tau^\pi(s, a)] - \tau h(\pi(s))$ which leads to the Bellman equation

$$Q_\tau^\pi(s, a) = R(s, a) + \gamma \mathbb{E}_{s', a'} [Q_\tau^\pi(s', a') - \tau h(\pi(s'))]. \tag{4}$$

In the following, we will write policies of the form $\pi(s) \propto \exp(Q(s, \cdot))$ for all $s \in S$ more succinctly as $\pi \propto \exp(Q)$. We define optimal V and Q functions where

$$\text{for all } s \in S, a \in A, V_\tau^\star(s) := \max_\pi V_\tau^\pi(s) \text{ and } Q_\tau^\star(s, a) := \max_\pi Q_\tau^\pi(s, a) \tag{5}$$

Moreover, the policy $\pi^\star \propto \exp\left(\frac{Q_\tau^\star}{\tau}\right)$ satisfies $Q_\tau^{\pi^\star} = Q_\tau^\star$ and $V_\tau^{\pi^\star} = V_\tau^\star$ simultaneously for all $s \in S$ (Zhan et al., 2023). In the following, we will overload notations of real functions defined on $S \times A$ and allow them to only take a state input and return a vector in $\mathbb{R}^{|A|}$. For example, $Q_\tau^\pi(s)$ denotes a vector for which the $i^{\text{th}}$ entry $i \in \{1, \ldots, |A|\}$ is equal to $Q_\tau^\pi(s, i)$.

For the convergence analysis, we will make use of a matrix representation of the MDP by overloading the notations of $R$ and $P$. Let $I_s : S \mapsto \{1, \ldots, |S|\}$ be an arbitrary bijective function that will provide an ordering over the state space, and we let $I_{sa} : S \times A \mapsto \{1, \ldots, |S||A|\}$ be an arbitrary bijective function that orders the state-action space. Using these indexing functions, we will overload the notation of the reward function by seeing $R$ as an $|S||A| \times 1$ matrix such that row $I_{sa}(s, a)$ and column 1 of the matrix $R$ verifies $R_{(I_{sa}(s,a),1)} = R(s, a)$. Similarly for the transition function $P$ which we see as an $|S||A| \times |S|$ matrix such that $P_{(I_{sa}(s,a),I_s(s'))} = P(s'|s, a)$.

A policy $\pi$ will be seen as a $|S| \times |S||A|$ matrix such that $\pi_{(I_s(s'),I_{sa}(s,a))} = \pi(a|s)$ if $s' = s$ and 0 otherwise. On such matrix representation of the policy we can apply the negative entropy row-wise such that $h(\pi)$ is a $|S| \times 1$ matrix where $h(\pi)_{(I_s(s),1)} = h(\pi(s))$. Using all of the above notations, we write the Bellman operator/equation associated to policy $\pi$ in matrix notation over a Q-function represented by an $|S||A| \times 1$ matrix that verifies

$$T_\tau^\pi Q_\tau^\pi = Q_\tau^\pi = R + \gamma P \left[\pi Q_\tau^\pi - \tau h(\pi)\right]. \tag{6}$$

This operator can be applied to any $|S||A| \times 1$ matrix $f$, by simply replacing $Q_\tau^\pi$ by $f$ in Eq. 6. We also write the Bellman optimality operator on matrices as

$$T_\tau^\star f = R + \gamma P \left[\max_p pf - \tau h(p)\right], \tag{7}$$

where the maximization $\max_p$ is made row-wise over probability matrices of shape $|S| \times |S||A|$ encoded using the same convention as policy matrices $\pi$ described above.

## 4 POLICY MIRROR DESCENT AND AVERAGING OF ERROR

To find $\pi^\star$, we focus on Entropy-regularized Policy Mirror Descent (EPMD) methods (Neu et al., 2017; Abbasi-Yadkori et al., 2019; Lazic et al., 2021) and notably on those that regularize the policy update with an entropy and $D_{\text{KL}}$ term and use an entropy regularized Bellman operator (Lan, 2022; Zhan et al., 2023). The EPMD setting discussed here is also similar to the regularized natural policy gradient algorithm on softmax policies of Cen et al. (2022). We will put special emphasis in this section on policy evaluation errors and show how convergence of EPMD depends on this error.

### 4.1 ENTROPY REGULARIZED VALUE ITERATION

To ease the discussion, let us first consider an approximate value iteration algorithm. Let $\xi_k : S \times A \mapsto \mathbb{R}$ be the unnormalized log-probability (which we refer to as logits for short) of $\pi_k$, i.e.

$$\pi_k \propto \exp(\xi_k). \tag{8}$$

We define entropy regularized value iteration by the following two steps.

**Evaluation step:** let $q_k : S \times A \mapsto \mathbb{R}$ be a sequence of functions such that $q_0 = 0$ and for all $k \geq 0$, $q_{k+1} = T_\tau^{k+1} q_k + \epsilon_{k+1}$, where $T_\tau^{k+1} = T_\tau^{\pi_{k+1}}$ is the Bellman operator associated to policy $\pi_{k+1}$ and $\epsilon_{k+1}$ represents the evaluation error due to, e.g., knowing only a sample estimate of the Bellman operator or knowing it only on a sub-set of the state-action space.

**Policy update step:** letting $\xi_k = 0$, i.e. the first policy is uniform over the action space, for each $q_k$ we update the policy in EPMD by solving the following optimization problem

$$\forall s \in S, \quad \pi_{k+1}(s) = \underset{p \in \Delta(A)}{\arg \max} \{q_k(s) \cdot p - \tau h(p) - \eta D_{\mathrm{KL}}(p; \pi_k(s))\} \tag{9}$$

where $D_{\mathrm{KL}}(p; p') = p \cdot (\log p - \log p')$ and $\eta > 0$ is the $D_{\mathrm{KL}}$ regularization weight. This update admits the well known closed-form solution given by

$$\xi_{k+1} = \beta \xi_k + \alpha q_k, \tag{10}$$

where $\alpha = \frac{1}{\eta + \tau}$ and $\beta = \frac{\eta}{\eta + \tau}$. Let us characterize the convergence of such an algorithm. In the remainder of this paper, we will be interested in bounding the norm $\|Q_\tau^\star - \tau \xi_k\|_\infty$ which we want as small as possible since the logits of the optimal policy are $\frac{Q_\tau^\star}{\tau}$. Moreover, from a bound over $\|Q_\tau^\star - \tau \xi_k\|_\infty$ we can derive the more common bound over Q-functions since

**Lemma 4.1.** *Let policy $\pi_k \propto \exp(\xi_k)$ and $Q_\tau^k$ its Q-function; then $\left\|Q_\tau^\star - Q_\tau^k\right\|_\infty \leq \frac{2\|Q_\tau^\star - \tau \xi_k\|_\infty}{1 - \gamma}$.*

The proof for all theoretical statements are given in the appendix. The convergence of entropy regularized value iteration is given by the following theorem

**Theorem 4.2.** *(Convergence of entropy regularized value iteration) Letting $E_j = (1 - \beta) \sum_{i=1}^j \beta^{j-i} \epsilon_i$ and $R_m = R_x + \gamma \tau \log |A|$, we have at iteration $k + 1$ that $\|Q_\tau^\star - \tau \xi_{k+1}\|_\infty \leq \gamma^{k+1} \|Q_\tau^\star\|_\infty + R_m \sum_{i=0}^k \gamma^i \beta^{k-i} + \sum_{i=0}^k \gamma^i \|E_{k-i}\|_\infty$.*

The first term of the upper bound goes to zero as $k \to \infty$. This term is also found in unregularized value iteration (see, e.g. Theorem 1.12 of Agarwal et al. (2021a)) and is due to the contraction property of the Bellman operators. The second term is a constant multiplied by $\sum_{i=0}^k \gamma^i \beta^{k-i}$. It can be shown that this sum satisfies

$$\sum_{i=0}^k \gamma^i \beta^{k-i} \leq \max\{\gamma, \beta\}^k (k + 1), \tag{11}$$

which goes to zero as $k \to \infty$. In the limit of $\eta \to 0$, where we would drop the $D_{\mathrm{KL}}$ regularization, $\beta \to 0$ and $\sum_{i=0}^k \gamma^i \beta^{k-i} \to \gamma^k$, yielding an error term that goes to zero at the same rate as $\gamma^{k+1} \|Q_\tau^\star\|_\infty$. However, as we increase the $D_{\mathrm{KL}}$ regularization, $\beta$ approaches 1 and this second term becomes whenever $\beta > \gamma$

$$\sum_{i=0}^k \gamma^i \beta^{k-i} = \beta^k \sum_{i=0}^k \left(\frac{\gamma}{\beta}\right)^i \leq \frac{\beta^{k+1}}{\beta - \gamma}. \tag{12}$$

While this term still goes to zero as $k \to \infty$, by increasing the $D_{\mathrm{KL}}$ regularization we pay the price of a slower convergence when $\beta > \gamma$. Finally, the term $\sum_{i=0}^k \gamma^i \|E_{k-i}\|_\infty$ constitutes the error floor of entropy regularized value iteration stemming from the evaluation errors $\epsilon_i$. This error floor might remain above zero even as $k \to \infty$. In the limit of $\eta \to 0$, $\|E_j\|_\infty \to \|\epsilon_j\|_\infty$ and the error floor will tend to $\sum_{i=0}^k \gamma^i \|\epsilon_j\|_\infty$, i.e. a weighted sum of the norms of the evaluation errors. However, as we increase the $D_{\mathrm{KL}}$ penalization, there is a hope that the evaluation errors will cancel each other in $(1 - \beta) \sum_{i=1}^j \beta^{j-i} \epsilon_i$, leading to a lower value of $\|E_j\|_\infty$ than if we would only consider the norm of the last error $\|\epsilon_j\|_\infty$. As such, by increasing $\eta$ we might slow down the convergence rate but potentially lower the error floor $\sum_{i=0}^k \gamma^i \|E_{k-i}\|_\infty$ and return a better final policy. This result is similar to that of Vieillard et al. (2020a), except that our algorithm uses a Bellman operator that only applies entropy regularization (as used for example in the learning of Q-functions in SAC (Haarnoja et al., 2018)), whereas Vieillard et al. (2020a) considered the Bellman operator that applies both an entropy and a $D_{\mathrm{KL}}$ regularization. Moreover, the latter work was restricted to the analysis of value iteration, but before discussing StaQ we need first to extend the above analysis to policy iteration as StaQ is a policy iteration algorithm.

## 4.2 ENTROPY REGULARIZED POLICY ITERATION

The policy iteration version of EPMD is quite similar to value iteration except that in the evaluation step, $q_k = Q_\tau^k + \epsilon_k$, where $Q_\tau^k = Q_\tau^{\pi_k}$ is the Q-function associated to $\pi_k$. The approximation $q_k$ of

$Q_\tau^k$ can be obtained for instance by applying the Bellman operator $T_\tau^k$ (or a noisy version thereof) several times on $q_{k-1}$—instead of a single time in value iteration. As value and policy iteration algorithms are quite similar, the analysis of the latter follows the same general template, except that the error propagates in slightly more complex ways. In the case of policy iteration, we need a way to relate $q_k$ and $q_{k+1}$ as this relation is not as direct as in value iteration. To do so, we will make use of the policy improvement lemma (Section 4.2 of Sutton & Barto (2018)). In the unregularized RL case, the lemma states that a policy greedy w.r.t. a Q-function will have a greater or equal Q-function at every state-action pair. A similar property holds in the entropy regularized case. However, in the presence of policy evaluation errors, the new policy $\pi_{k+1}$—obtained by maximizing Eq. 9—might increase the probability of sub-par actions and improvement is only guaranteed up to some policy improvement error characterized by the following lemma

**Lemma 4.3** (Approximate policy improvement). *Let $\mu_k = (I - \gamma P \pi_k)^{-1}$ be the (unnormalized) state distribution associated to policy $\pi_k$. At any iteration $k \geq 0$ of entropy regularized policy iteration, we have that $Q_\tau^{k+1} \geq Q_\tau^k - \epsilon_{\Delta_k}$, with $\epsilon_{\Delta_k} := \gamma \mu_{k+1} P(\pi_k - \pi_{k+1}) \epsilon_k$.*

The policy improvement error $\epsilon_{\Delta_k}$ is invariant to a constant shift in the evaluation error $\epsilon_k$. Indeed, we have that for any real value $c$, $(\pi_k - \pi_{k+1})(\epsilon_k + c\mathbf{1}) = (\pi_k - \pi_{k+1})\epsilon_k$, where $\mathbf{1}$ is a vector of ones. Additionally, if $\epsilon_k = 0$ or any other constant vector then policy improvement is guaranteed. However, we might not improve over the previous Q-function if we overestimate a bad action or underestimate a good one. The overall convergence of entropy regularized policy iteration is characterized by the following theorem

**Theorem 4.4.** *(Convergence of entropy regularized policy iteration) Letting $\epsilon_{-1} = 0$ and $E_j := (1 - \beta) \sum_{i=0}^{j} \beta^{j-i}(\epsilon_i - \gamma P \pi_i(\epsilon_{i-1} + \epsilon_{\Delta_i}))$, we have at iteration $k + 1$ that $\|Q_\tau^\star - \tau \xi_{k+1}\|_\infty \leq \gamma^{k+1} \|Q_\tau^\star\|_\infty + \frac{2-\gamma-\beta}{1-\gamma} R_m \sum_{i=0}^{k} \gamma^i \beta^{k-i} + \sum_{i=0}^{k} \gamma^i \|E_{k-i}\|_\infty.$*

As can be seen, the upper bound of Theorem 4.4 follows a very similar structure to that of value iteration, with the main difference being in the error floor $\sum_{i=0}^{k} \gamma^i \|E_{k-i}\|_\infty$ that now notably depends on the policy improvement error discussed above. While this error floor involves more quantities, the general scheme remains the same and one hopes that there are values of $\eta$ such that a cancellation of terms leads to a lower error floor compared to the unregularized case while not slowing policy iteration too much. This analysis improves over that of Zhan et al. (2023), that only considered a uniform worst case error, leading to a less interesting upper bound where the smallest error floor—and the fastest convergence rate—is always obtained by choosing $\eta = 0$, using no $D_{\mathrm{KL}}$ regularization.

### 4.3 APPROXIMATE POLICY UPDATE

We have analyzed so far the convergence of EPMD algorithms, considering only evaluation errors. In practice, the policy update step that consists in solving the optimization problem in Eq. 9 might prove challenging to solve without approximations. Indeed, while this policy update leads to a closed form solution in the space of policy logits (Eq. 10), it might not be possible to implement exactly if the state-action space is too large. In this case, one could use a function approximator to represent these logits but that would likely introduce a new type of error and raises the question of what loss to use to update the policy's parameters.

Let $\Theta_Q$ and $\Theta_\pi$ respectively be the parameter spaces of Q-functions and policy logits; these parameter spaces can for instance be subsets of $\mathbb{R}^d$ for some integer $d$. Let $\xi_\theta : S \times A \mapsto \mathbb{R}$, with $\theta \in \Theta_\pi$ be a function that provides the logits of a policy $\pi_\theta \propto \exp(\xi_\theta)$ and let $q_{\theta'} : S \times A \mapsto \mathbb{R}$, with $\theta' \in \Theta_\pi$, be the (approximate) Q-function associated to $\pi_\theta$. We want to find $\xi_{\theta''}$ as the solution to the entropy regularized policy update in Eq. 9. We know that ideally we would have $\xi_{\theta''} = \beta \xi_\theta + \alpha q_{\theta'}$, but this does not give an expression for $\theta''$ since in general the policy maximizing Eq. 9 is not necessarily parameterized by $\beta\theta + \alpha\theta'$.

One exception to the above claim is when the Q-function and the logits function are linear w.r.t. some predefined feature function, in which case $\xi_{\beta\theta+\alpha\theta'} = \beta \xi_\theta + \alpha q_{\theta'}$. This is the so-called compatible feature setting of policy gradient (Sutton et al., 1999; Geist & Pietquin, 2010; Pajarinen et al., 2019) where the Q-function and the logits share the same linear-in-feature function approximation class and in which case, policy gradient and natural policy gradient become equivalent (Peters & Schaal, 2008). However, beyond the linear-in-feature case, the closed form solution of the entropy

regularized policy update in the space of logits does not yield a trivial update in parameter space.

One approach to policy update in parameter space is to solve the following optimization problem

$$\underset{\theta'' \in \Theta_\pi}{\arg\max} \; \theta'' \cdot \nabla_\theta J_\tau(\pi_\theta) - \eta \mathbb{E}_{s \sim \mu_{\pi_\theta}} D_{\mathrm{KL}}(\pi_{\theta''}(s); \pi_\theta(s)), \tag{13}$$

where $J_\tau(\pi_\theta)$ is the policy return $\pi_\theta$, i.e. the expectation of the $V^{\pi_\theta}$ over states sampled from a predefined initial state distribution. Interestingly, Cen et al. (2022) showed that in the tabular case (i.e. when optimizing directly over the logit space), the above problem is equivalent to the maximization in Eq. 9 over each state independently. In the general case however, because of the approximation in the optimization of Eq. 13 or because of the use of a restricted policy class, we are likely to obtain a new policy $\pi_{\theta''}$ that is worse than $\pi_{k+1}$ in the sense of the policy update objective defined in Eq. 9.

In Zhan et al. (2023), the authors analyzed an approximate EPMD scheme such that the policy update objective in Eq. 9 is optimized up to some error $\epsilon_{\mathrm{opt}}$ for all states and all iterations. They showed that the resulting algorithm would converge at the same rate as its exact counterpart but would reach an error floor that depends on $\epsilon_{\mathrm{opt}}$, independently of the existence of policy evaluation errors. In this paper, we investigate an alternative policy update, that truncates the infinite sum of EPMD to result in a practical algorithm that does not introduce errors from the policy update yet keeps the appealing property of averaging policy evaluation errors.

## 5 FINITE-MEMORY POLICY MIRROR DESCENT

Let us now consider a PMD-like algorithm that keeps in memory at most $M$—with $M$ being a finite and strictly positive integer—Q-function estimates. The policy at iteration $k$ is now given by Eq. 2, which can be written as a recursive update in the logits space in the following way

$$\xi_{k+1} = \beta \xi_k + \alpha q_k + \frac{\alpha \beta^M}{1 - \beta^M}(q_k - q_{k-M}), \tag{14}$$

where $q_{k-M} := 0$ whenever $k - M < 0$, and $q_k = Q_\tau^k + \epsilon_k$ otherwise. In contrast to vanilla EPMD in Eq. 10, we now 'delete' at each update the oldest Q-function estimate $q_{k-M}$ and also slightly overweight the most recent Q-function estimate to ensure that the Q-function weights sum to 1. This weight correction in Eq. 2—the extra multiplication by $\frac{1}{1-\beta^M}$ compared to the vanilla sum in Eq. 1—is important as otherwise, the logits might never converge to $\frac{Q_\tau^\star}{\tau}$ even when the last $M$ Q-function estimates are all equal to $Q_\tau^\star$. Indeed, without the weight correction and since $\tau\alpha = 1 - \beta$, we would have $\tau\xi_k = (1 - \beta)\sum_{i=0}^{M-1}\beta^i Q_\tau^\star = (1 - \beta^M)Q_\tau^\star$.

The logits update in Eq. 14 can be interpreted as the result of the following optimization problem

$$\forall s \in S, \pi_{k+1}(s) = \underset{p \in \Delta(A)}{\arg\max} \; p \cdot [q_k + \frac{\beta^M}{1 - \beta^M}(q_k - q_{k-M})](s) - \tau h(p) - \eta D_{\mathrm{KL}}(p; \pi_k(s)) \tag{15}$$

Now instead of maximizing the latest Q-function estimate, the policy also maximizes the difference between the latest and oldest estimate out of the last $M$ Q-functions. This introduces an additional source of policy improvement error, but in our theoretical analysis, we show that for a finite but large enough $M$, this error will vanish as $k \to \infty$, leaving us with an error floor that only depends on the evaluation errors. Specifically, the algorithm given by Eq. 14 has the following convergence properties

**Theorem 5.1** (Convergence of finite memory EPMD). *Let $M$ such that $M > \log\frac{(1-\gamma)^3}{(1+\gamma)(1-\gamma)^2+4(\gamma+\gamma^2)}/\log\beta$, $\gamma_M = \frac{\gamma}{1-\beta^M}$, $c = \frac{\beta^M}{1-\beta^M}\left(\frac{4(\gamma+\gamma^2)}{(1-\gamma)^2} + \gamma\right)$, let $d_0$ be the unique root of $d^{2M+1} - \gamma_M d^{2M} - c$ in the interval $(\gamma_M, 1)$, define the matrix $A_{k+1} = \gamma P\pi_{k+1}(I + \gamma\mu_{k+1}P(\pi_{k+1} - \pi_k))$, and error terms $E_j = \frac{1-\beta}{1-\beta^M}\sum_{i=0}^{M-1}\beta^i(\epsilon_{k-i} - A_{k-i}\epsilon_{k-i-1})$, and $T_k = \|E_k\|_\infty + \frac{\beta^M(1-\beta)}{(1-\beta^M)^2}\left\|\sum_{i=0}^{M-1}\beta^i A_{k-i}(\epsilon_{k-i-1} - \epsilon_{k-i-1-M})\right\|_\infty$ and worst error term $\bar{T} = \max_{0 \le i \le k} T_i$, then $\|Q_\tau^\star - \xi_{k+1}\|_\infty \le d_0^{k+1}\|Q_\tau^\star\|_\infty + \sum_{i=0}^k \gamma_M^i\left(T_{k-i} + c\frac{\bar{T}}{1-\gamma_M-c}\right).$*

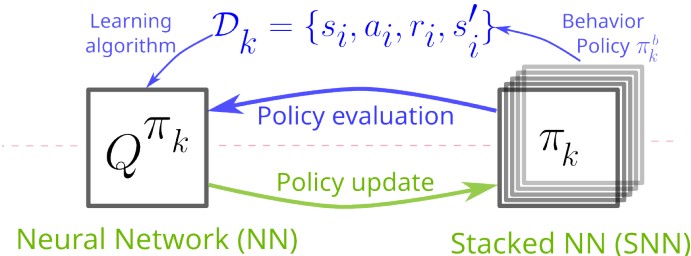

Figure 1: Overview of StaQ, showing the continual training of a Q-function (left), from which we periodically "stack" frozen weight snapshots to form the policy (right). See Sec. 5.1 for more details. At each iteration $k$, two steps are performed. i) Policy evaluation, where we generate a dataset $\mathcal{D}_k$ of transitions that are gathered by a behavior policy $\pi_k^b$, typically derived from $\pi_k$, and then learn $Q^{\pi_k}$ from $\mathcal{D}_k$. ii) Policy update, performed by "stacking" the NN of $Q^{\pi_k}$ into the current policy. The policy update is optimization-free and theoretically grounded (Sec. 5), thus only the choice of $\pi_k^b$ and the policy evaluation algorithm can remain sources of instabilities in this deep RL setting.

In finite memory EPMD we no longer have an explicit expression for the convergence rate, but provided $M$ is large enough, we know that it exists as the unique root of a function in the range $(\gamma_M, 1)$. Moreover, as $M \to \infty$, the convergence rate goes to $\gamma$, and as in exact EPMD, we note that in the absence of evaluation errors, the algorithm converges to the optimal policy. In terms of error floor, we find a similar expression of $E_j$ as in Sec. 4.2 up to the truncation to the latest $M$ errors. However, the error floor does not depend only on $E_j$ but on an additional term in $T_j$ that depends on older evaluation error terms. This additional term is weighted by $\frac{\beta^M(1-\beta)}{(1-\beta^M)^2}$ which decreases exponentially fast towards zero as we increase $M$. Finally, the error floor is not just a weighted sum of average evaluation errors but also depends on the worst average evaluation error $\bar{T}$. Fortunately, this term is weighted again by a constant that decreases exponentially fast towards zero as $M$ increases. Thus the theoretical analysis indicates in terms of averaging of policy evaluation errors, we approach the desirable behavior of an exact EPMD policy update exponentially fast by increasing $M$.

The remainder of the paper is devoted to exploring the practical implications of such a result: what are the tangible benefits of EPMD and how fast—in terms of memory size $M$—can a finite memory variant approach the behavior of EPMD with an exact policy update. To perform such a study we need an efficient implementation of EPMD that allows us to examine exact EPMD, at least for a few million timesteps, and support a high memory size $M$.

## 5.1 PRACTICAL IMPLEMENTATION

We implement an efficient version of the policy in Eq. 2 using a stacked neural networks (SNN, illustrated in Fig. 1). By using batched operations we make efficient use of GPUs and compute multiple Q-values in parallel. We call the resulting algorithm StaQ. After each policy evaluation, we push the weights corresponding to this new Q-function onto the stack. If the stacked NN contains more than $M$ NNs, the oldest NN is deleted in a "first in first out" fashion. If implementing exact EPMD, then we never delete older Q-functions.

To further reduce the impact of a large $M$, we pre-compute $\xi_k$ for all entries in the replay buffer[1] at the start of policy evaluation. The logits $\xi_k$ are used to sample on-policy actions when computing the targets for $Q_\tau^k$. As a result of the pre-computation, during policy evaluation, forward and backward passes only operate on the current Q-function and hence the impact of large $M$ is minimized. However rolling out the current behavioural policy $\pi_k^b$ still requires a full forward pass. Conversely, the policy update consists only of adding the new weights to the stack, and thus, is optimization free and (almost) instantaneous. Table 1 shows the training time of StaQ as a function of $M$ for two environments. Varying $M$ or the state space size has little impact on the runtime of StaQ on GPU, at least for these medium-sized environments.

---

[1]Since we use small replay buffer sizes of 50K transitions, we are likely to process each transition multiple times (25.6 times in expectation in our experiments) making this optimization worthwhile.

Table 1: Training times for StaQ (5 million steps), as a function of $M$, on Hopper-v4 (state dim.=11) and Ant-v4 (state dim. = 105), computed on an NVIDIA Tesla V100 and averaged over 3 seeds.

| | Memory size $M$ | 1 | 50 | 100 | 300 | 500 |
|---|---|---|---|---|---|---|
| Hopper-v4 | Training time (hrs) | 9.8 | 10.1 | 10.3 | 10.3 | 10.9 |
| Ant-v4 | Training time (hrs) | 10.4 | 10.7 | 10.3 | 11 | 10.5 |

## 6 EXPERIMENTS

**Environments.** We use all 9 environments suggested by Ceron & Castro (2021) for comparing deep RL algorithms with finite action spaces, comprising 4 classic control tasks from Gymnasium (Towers et al., 2023), and all MinAtar tasks (Young & Tian, 2019). To that we add 5 MuJoCo tasks (Todorov et al., 2012), adapted to discrete action spaces by considering only extreme actions similarly to (Seyde et al., 2021). To illustrate, the discrete version of a MuJoCo task with action space $A = [-1, 1]^d$ consists in several $2d$ dimensional vectors that have zeroes everywhere except at entry $i \in \{1, \ldots, d\}$ that can either take a value of 1 or $-1$; to that we add the zero action, for a total of $2d + 1$ actions.

**Algorithms.** In our main set of experiments, we compare finite memory EPMD for several values of $M$ on up to 5 million timesteps. We notably consider the two extremes of $M = 1$, using no $D_{\mathrm{KL}}$ regularization and $M = 1000$ that never deletes a Q-function within the 5 million timesteps window (labeled "Exact PMD" in figures). As lower values of $M$ such as $M = 1$ might decrease entropy too quickly because of too aggressive a $D_{\mathrm{KL}}$ between successive iterations, we add a constant probability $\epsilon = 0.05$ of sampling a random action throughout the learning phase for all algorithms. This reduces the differences between the algorithms on components outside of the error floor which is the main purpose of this experiment. To variants of StaQ, we consider another baseline that implements a policy update by approximately solving Eq. 13 using TRPO's conjugate gradient implementation. This baseline, labeled "NatGrad" uses the exact same policy evaluation procedure and hyperparameters (including regularization weights $\tau$ and $\eta$) as StaQ variants and differs only in the policy update.

Following common practices in natural gradient descent implementations, we also consider adding a post-update line search step that constrains the $D_{\mathrm{KL}}$ between successive policies to be under an $\epsilon_{\mathrm{KL}}$ threshold. To select an appropriate value for $\epsilon_{\mathrm{KL}}$, we performed a preliminary study in App. B.2, comparing the default value $\epsilon_{\mathrm{KL}} = 0.01$ used in many of TRPO's implementation—e.g. in `stable-baselines3` (Raffin et al., 2021)—with a smaller value of $\epsilon_{\mathrm{KL}} = 0.001$ which we found to work better in most of the tasks. This second baseline is labeled "NatGrad + LS" and has subtle differences with TRPO: for instance even when an entropy bonus is used, TRPO does not learn soft value functions and only regularizes the policy update with an additional entropy term. We found that by matching StaQ's setting, "NatGrad + LS" is comparable in performance to `stable-baselines3`'s TRPO on most of the tasks but can largely outperform it on others, especially on MinAtar tasks, as seen from the results in App. B.3.

In the appendix, we complement this experiment that isolates the policy update from the rest of the deep RL components, with a secondary set of comparisons to existing deep RL baselines such as the value iteration algorithm DQN (Mnih et al., 2015) and its entropy-regularized variant M-DQN (Vieillard et al., 2020b), the policy gradient algorithm TRPO (Schulman et al., 2015) and PPO (Schulman et al., 2017). StaQ performs entropy regularization on top of a Fitted-Q Iteration (FQI) approach. DQN only uses FQI and is a good baseline to measure the impact of entropy regularization over vanilla FQI, while the other baselines cover a wide range of alternative approaches to regularization in deep RL: through a bonus term (M-DQN), following the natural gradient (TRPO) or with a clipping loss (PPO). These baselines differ more widely and in ways orthogonal (policy evaluation, exploration, replay buffer management) to the main focus of this paper which is the policy update of EPMD. These results are thus harder to interpret and are only provided for reference.

**Metrics.** We launch 20 independent runs for each algorithm and each task and report normalized aggregated performance metrics. Normalization is performed by dividing final policy performance by the highest final policy evaluation across all algorithms and seeds. Final policy performance—and

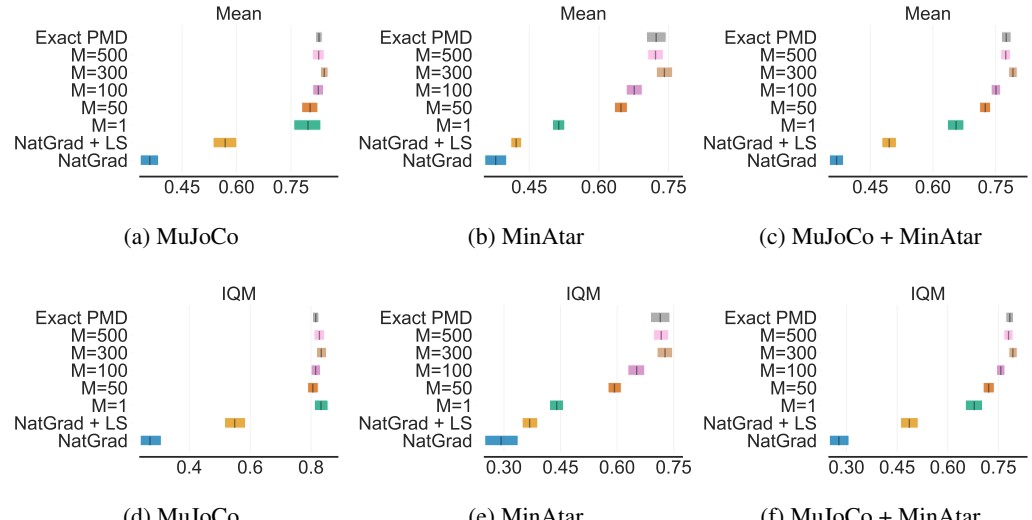

(a) MuJoCo         (b) MinAtar         (c) MuJoCo + MinAtar

(d) MuJoCo         (e) MinAtar         (f) MuJoCo + MinAtar

Figure 2: Normalized aggregate policy performance on 5 MuJoCo and 5 MinAtar tasks with 20 independent runs for each algorithm and task, showing mean and interquantile mean with 95% stratified bootstrap confidence interval. There is an almost monotonic increase in performance as a function of $M$, with $M \geq 100$ matching the mean performance of exact PMD on MuJoCo tasks, and $M \geq 300$ matching the mean performance of exact PMD on MinAtar tasks. Per task learning curves and box plots of final performances can be found in App. B.1.

intermediary policy evaluations reported in the appendix—are obtained by averaging the cumulative undiscounted rewards over 25 rollouts. Once normalized, policy performance is aggregated using the mean or interquantile mean with 95% stratified bootstrap confidence intervals as recommended by (Agarwal et al., 2021b). In the appendix, we provide training curves showing mean and 95% confidence intervals of the current policy evaluated every 200K environment steps, as well as per-task box plots of the final policy performance for each of the 20 runs of each algorithm.

**Results.** Fig 2 shows an almost monotonic improvement in performance on the aggregate plot, which is especially true for MinAtar tasks (both mean and IQM metrics) and for the mean performance on MuJoCo. Looking at per task performance in App. B.1, we find that for almost all environments, a sufficiently large $M$ matches the performance of exact EPMD, with $M \geq 300$ being virtually indistinguishable from exact EPMD on all tasks. These results reinforce the theoretical insights that StaQ can match the behavior of exact EPMD. Compared to natural policy gradient, the performance is generally improved across all tasks, indicating that StaQ is potentially a better alternative to the latter PMD approximate scheme, at least on environments where Q-function computations can be efficiently batched. When compared to deep RL baselines Fig. 6, we also noted lower inter-seed oscillations in StaQ, which we demonstrate explicitly in App. B.4.

# 7 CONCLUSION

In this paper, we proposed a policy update rule based on policy mirror descent, that keeps in memory at most $M$ Q-functions. We showed that by increasing $M$ we can quickly mimic exact EPMD both from a theoretical and empirical perspective. Surprisingly, even when $M$ is large, the final computational burden is small on modern hardware, due to the stacking of the Q-functions. The resulting policy update has a solid theoretical foundation and clear empirical benefits as it improves performance and reduces learning instability compared to other entropy regularization methods in the literature, making it a valid alternative to existing EPMD schemes, at least for medium-sized tasks such as MuJoCo or MinAtar. Due to its exact policy update, and the absence of gap between the theoretical algorithm and the practical implementation, StaQ provides a promising setting for testing other components of RL such as policy evaluation, for instance, the recent methods that use normalization techniques to reduce policy evaluation error (Gallici et al., 2025; Bhatt et al., 2024).

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

APPENDIX

# A PROOFS

This section includes proofs of the lemmas and theorems of the main paper.

## A.1 PROPERTIES OF ENTROPY REGULARIZED BELLMAN OPERATORS

We first start with a reminder of some basic properties of the (entropy regularized) Bellman operators, as presented in (Geist et al., 2019; Zhan et al., 2023). Within the MDP setting defined in Sec. 3, let $T_\tau^\pi$ be the operator defined for any map $f : S \times A \mapsto \mathbb{R}$ by

$$(T_\tau^\pi f)(s,a) = R(s,a) + \gamma \mathbb{E}_{s',a'}[f(s',a') - \tau h(\pi(s'))], \tag{16}$$

This operator has the following three properties.

**Proposition A.1** (Contraction). *$T_\tau^\pi$ is a $\gamma$-contraction w.r.t. the $\|.\|_\infty$ norm, i.e. $\|T_\tau^\pi f - T_\tau^\pi g\|_\infty \leq \gamma \|f - g\|_\infty$ for any real functions $f$ and $g$ of $S \times A$.*

**Proposition A.2** (Fixed point). *$Q_\tau^\pi$ is the unique fixed point of the operator $T_\tau^\pi$, i.e. $T_\tau^\pi Q_\tau^\pi = Q_\tau^\pi$.*

Let $f, g$ be two real functions of $S \times A$. We say that $f \geq g$ iff $f(s,a) \geq g(s,a)$ for all $(s,a) \in S \times A$.

**Proposition A.3** (Monotonicity). *$T_\tau^\pi$ is monotonous, i.e. if $f \geq g$ then $T_\tau^\pi f \geq T_\tau^\pi g$.*

Let the Bellman optimality $T_\tau^\star$ operator be defined by

$$(T_\tau^\star f)(s,a) = R(s,a) + \gamma \mathbb{E}_{s'} \left[ \max_{p \in \Delta(A)} f(s') \cdot p - \tau h(p) \right]. \tag{17}$$

For the Bellman optimality operator we need the following two properties.

**Proposition A.4** (Contraction). *$T_\tau^\star$ is a $\gamma$-contraction w.r.t. the $\|.\|_\infty$ norm, i.e. $\|T_\tau^\star f - T_\tau^\star g\|_\infty \leq \gamma \|f - g\|_\infty$ for any real functions $f$ and $g$ of $S \times A$.*

**Proposition A.5** (Optimal fixed point). *$T_\tau^\star$ admits $Q_\tau^\star$ as a unique fixed point, satisfying $T_\tau^\star Q_\tau^\star = Q_\tau^\star$.*

Finally, we will make use of the well known property that the softmax distribution is entropy maximizing (Geist et al., 2019). Specifically, we know that the policy $\pi_k \propto \exp(\xi_k)$ satisfies the following property

$$\text{for all } s \in S, \quad \pi_k(s) = \arg\max_{p \in \Delta(A)} \xi_k(s) \cdot p - h(p). \tag{18}$$

## A.2 PROOF OF LEMMA 4.1

*Proof.* We first observe from the definition of $\pi_k$ that

$$T_\tau^k \tau \xi_k = R + \gamma P(\pi_k \tau \xi_k - \tau h(\pi_k)), \tag{19}$$

$$\overset{(i)}{=} R + \gamma P(\max_p p \tau \xi_k - \tau h(p)), \tag{20}$$

$$= T_\tau^\star \tau \xi_k, \tag{21}$$

with $(i)$ due to Eq. 18. Then

$$Q_\tau^k - \tau\xi_k = (T_\tau^k Q_\tau^k - T_\tau^k \tau\xi_k) + (T_\tau^k \tau\xi_k - \tau\xi_k) \tag{22}$$

$$= \gamma P_k(Q_\tau^k - \tau\xi_k) + (T_\tau^k \tau\xi_k - \tau\xi_k), \tag{23}$$

$$= (I - \gamma P_k)^{-1}(T_\tau^k \tau\xi_k - \tau\xi_k), \tag{24}$$

$$= (I - \gamma P_k)^{-1}(T_\tau^\star \tau\xi_k - \tau\xi_k), \tag{25}$$

$$\Rightarrow \left\| Q_\tau^k - \tau\xi_k \right\|_\infty \le \frac{1}{1-\gamma} \left\| T_\tau^\star \tau\xi_k - \tau\xi_k \right\|_\infty, \tag{26}$$

$$= \frac{1}{1-\gamma} \left\| T_\tau^\star \tau\xi_k - Q_\tau^\star + Q_\tau^\star - \tau\xi_k \right\|_\infty, \tag{27}$$

$$\le \frac{1}{1-\gamma} \left( \left\| T_\tau^\star Q_\tau^\star - T_\tau^\star \tau\xi_k \right\|_\infty + \left\| Q_\tau^\star - \tau\xi_k \right\|_\infty \right), \tag{28}$$

$$\le \frac{1+\gamma}{1-\gamma} \left\| Q_\tau^\star - \tau\xi_k \right\|_\infty. \tag{29}$$

Finally,

$$\left\| Q_\tau^\star - Q_\tau^k \right\|_\infty \le \left\| Q_\tau^\star - \tau\xi_k \right\|_\infty + \left\| Q_\tau^k - \tau\xi_k \right\|_\infty, \tag{30}$$

$$\le \left\| Q_\tau^\star - \tau\xi_k \right\|_\infty + \frac{1+\gamma}{1-\gamma} \left\| Q_\tau^\star - \tau\xi_k \right\|_\infty, \tag{31}$$

$$= \frac{2 \left\| Q_\tau^\star - \tau\xi_k \right\|_\infty}{1-\gamma}. \tag{32}$$

$\square$

## A.3 PROOF OF THEOREM 4.2

*Proof.* Looking at the value function, we have

$$\pi_{k+1} q_k - \tau h(\pi_{k+1}) = \pi_{k+1}\left( \frac{1}{1-\beta}(\tau\xi_{k+1} - \beta\tau\xi_k) \right) - \tau h(\pi_{k+1}), \tag{33}$$

$$= \pi_{k+1}\left( \frac{1}{1-\beta}(\tau\xi_{k+1} - \beta\tau\xi_k) \right) - \frac{1}{1-\beta}(\tau h(\pi_{k+1}) - \beta\tau h(\pi_{k+1})), \tag{34}$$

$$= \pi_{k+1}\left( \frac{1}{1-\beta}[\tau\xi_{k+1} - \tau h(\pi_{k+1}) - \beta(\tau\xi_k - \tau h(\pi_{k+1}))] \right), \tag{35}$$

$$\stackrel{(i)}{\ge} \left( \frac{1}{1-\beta}[\pi_{k+1}\tau\xi_{k+1} - \tau h(\pi_{k+1}) - \beta(\pi_k \tau\xi_k - \tau h(\pi_k))] \right), \tag{36}$$

with $(i)$ due to $\pi_k \tau\xi_k - \tau h(\pi_k) = \max_p p\tau\xi_k - \tau h(p) \ge \pi_{k+1}\tau\xi_{k+1} - \tau h(\pi_{k+1})$. Using this inequality in $q_{k+1}$ yields

$$q_{k+1} = T_\tau^{k+1} q_k + \epsilon_{k+1}, \tag{37}$$

$$= R + \gamma P(\pi_{k+1} q_k - \tau h(\pi_{k+1})) + \epsilon_{k+1}, \tag{38}$$

$$\ge R + \gamma P\left( \frac{1}{1-\beta}[\pi_{k+1}\tau\xi_{k+1} - \tau h(\pi_{k+1}) - \beta(\pi_k \tau\xi_k - \tau h(\pi_k))] \right) + \epsilon_{k+1}, \tag{39}$$

$$= \frac{1}{1-\beta}(R - \beta R) + \gamma P\left( \frac{1}{1-\beta}[\pi_{k+1}\tau\xi_{k+1} - \tau h(\pi_{k+1}) - \beta(\pi_k \tau\xi_k - \tau h(\pi_k))] \right) + \epsilon_{k+1}, \tag{40}$$

$$= \frac{1}{1-\beta}(T_\tau^{k+1}\tau\xi_{k+1} - \beta T_\tau^k \tau\xi_k) + \epsilon_{k+1}, \tag{41}$$

$$= \frac{1}{1-\beta}(T_\tau^\star \tau\xi_{k+1} - \beta T_\tau^\star \tau\xi_k) + \epsilon_{k+1}, \tag{42}$$

where the last step is again due to Eq. 18. Now using this inequality in the definition of $\xi_{k+1}$ gives

$$\tau\xi_{k+1} = (1-\beta)\sum_{i=0}^{k}\beta^{k-i}q_i, \tag{43}$$

$$\stackrel{(i)}{=} (1-\beta)\sum_{i=1}^{k}\beta^{k-i}q_i, \tag{44}$$

$$\geq (1-\beta)\sum_{i=1}^{k}\beta^{k-i}\left(\frac{1}{1-\beta}(T_\tau^\star\tau\xi_i - \beta T_\tau^\star\tau\xi_{i-1}) + \epsilon_i\right), \tag{45}$$

$$= T_\tau^\star\tau\xi_k - \beta^k T_\tau^\star\tau\xi_0 + (1-\beta)\sum_{i=1}^{k}\beta^{k-i}\epsilon_i, \tag{46}$$

with (i) due to $q_0 = 0$. Letting $E_j = (1-\beta)\sum_{i=1}^{j}\beta^{j-i}\epsilon_i$ and $R_m = R_x + \gamma\tau\log|A|$ be an upper bound to $\|T_\tau^\star\tau\xi_0\|_\infty$, we finally obtain

$$Q_\tau^\star - \tau\xi_{k+1} \leq Q_\tau^\star - T_\tau^\star\tau\xi_k + \beta^k T_\tau^\star\tau\xi_0 - E_k, \tag{47}$$

$$\Rightarrow \|Q_\tau^\star - \tau\xi_{k+1}\|_\infty \leq \|Q_\tau^\star - T_\tau^\star\tau\xi_k\|_\infty + \beta^k\|T_\tau^\star\tau\xi_0\|_\infty + \|E_k\|_\infty, \tag{48}$$

$$\leq \gamma\|Q_\tau^\star - \tau\xi_k\|_\infty + \beta^k R_m + \|E_k\|_\infty, \tag{49}$$

$$\leq \gamma^{k+1}\|Q_\tau^\star\|_\infty + R_m\sum_{i=0}^{k}\gamma^i\beta^{k-i} + \sum_{i=0}^{k}\gamma^i\|E_{k-i}\|_\infty. \tag{50}$$

if $\beta > \gamma$

$$\sum_{i=0}^{k}\gamma^i\beta^{k-i} = \beta^k\sum_{i=0}^{k}\left(\frac{\gamma}{\beta}\right)^i, \tag{51}$$

$$= \frac{\beta^{k+1} - \gamma^{k+1}}{\beta - \gamma}. \tag{52}$$

if $\beta < \gamma$

$$\sum_{i=0}^{k}\gamma^i\beta^{k-i} = \sum_{i=0}^{k}\gamma^{k-i}\beta^i, \tag{53}$$

$$= \frac{\gamma^{k+1} - \beta^{k+1}}{\gamma - \beta}. \tag{54}$$

if $\beta = \gamma$

$$\sum_{i=0}^{k}\gamma^i\beta^{k-i} = \gamma^k(k+1). \tag{55}$$

In all cases

$$\sum_{i=0}^{k}\gamma^i\beta^{k-i} \leq \max\{\gamma,\beta\}^k(k+1). \tag{56}$$

$\square$

### A.4 Proof of Lemma 4.3

*Proof.* As $\pi_{k+1}$ maximizes the policy update Eq. 9, and from the non-negativity of the $D_{\mathrm{KL}}$ and the fact that $D_{\mathrm{KL}}(\pi_k;\pi_k) = 0$ we have

$$\pi_k q_k - \tau h(\pi_k) \leq \pi_{k+1}q_k - \tau h(\pi_{k+1}) - \eta D_{\mathrm{KL}}(\pi_{k+1};\pi_k), \tag{57}$$

$$\leq \pi_{k+1}q_k - \tau h(\pi_{k+1}), \tag{58}$$

$$\Leftrightarrow \pi_k Q_\tau^k - \tau h(\pi_k) \leq \pi_{k+1}Q_\tau^k - \tau h(\pi_{k+1}) + (\pi_{k+1} - \pi_k)\epsilon_k. \tag{59}$$

$$Q_\tau^{k+1} - Q_\tau^k = \gamma P(\pi_{k+1} Q_\tau^{k+1} - \tau h(\pi_{k+1})) - \gamma P(\pi_k Q_\tau^k - \tau h(\pi_k)), \qquad (60)$$

$$\overset{(i)}{\geq} \gamma P(\pi_{k+1} Q_\tau^{k+1} - \tau h(\pi_{k+1})) - \gamma P(\pi_{k+1} Q_\tau^k - \tau h(\pi_{k+1}) + (\pi_{k+1} - \pi_k)\epsilon_k), \qquad (61)$$

$$= \gamma P \pi_{k+1}(Q_\tau^{k+1} - Q_\tau^k) + \gamma P(\pi_k - \pi_{k+1})\epsilon_k, \qquad (62)$$

$$\Leftrightarrow (I - \gamma P \pi_{k+1})(Q_\tau^{k+1} - Q_\tau^k) \geq \gamma P(\pi_k - \pi_{k+1})\epsilon_k, \qquad (63)$$

$$\Leftrightarrow Q_\tau^{k+1} - Q_\tau^k \overset{(ii)}{\geq} \gamma(I - \gamma P \pi_{k+1})^{-1} P(\pi_k - \pi_{k+1})\epsilon_k. \qquad (64)$$

where in $(i)$ we have used the fact that $P$ is a probability matrix with only positive entries, and similarly in $(ii)$ for the matrix $(I - \gamma P \pi_{k+1})^{-1} = \sum_{i=0}^\infty (\gamma P \pi_{k+1})^i$. $\qquad \square$

## A.5 PROOF OF THEOREM 4.4

*Proof.* The beginning of the proof is the same as in value iteration and we can show using the same arguments that

$$\pi_{k+1} q_k - \tau h(\pi_{k+1}) \geq \left( \frac{1}{1-\beta} [\pi_{k+1}\tau\xi_{k+1} - \tau h(\pi_{k+1}) - \beta(\pi_k \tau \xi_k - \tau h(\pi_k))] \right). \qquad (65)$$

Using this inequality in $q_{k+1}$ yields

$$q_{k+1} = Q_\tau^{k+1} + \epsilon_{k+1}, \qquad (66)$$

$$= R + \gamma P(\pi_{k+1} Q_\tau^{k+1} - \tau h(\pi_{k+1})) + \epsilon_{k+1}, \qquad (67)$$

$$\overset{(i)}{\geq} R + \gamma P(\pi_{k+1}(q_k - \epsilon_k - \epsilon_{\Delta_k}) - \tau h(\pi_{k+1})) + \epsilon_{k+1}, \qquad (68)$$

$$\geq R + \gamma P \left( \frac{1}{1-\beta}[\pi_{k+1}\tau\xi_{k+1} - \tau h(\pi_{k+1}) - \beta(\pi_k \tau\xi_k - \tau h(\pi_k))] \right) + \epsilon_{k+1} - \gamma P_{k+1}(\epsilon_k + \epsilon_{\Delta_k}), \qquad (69)$$

$$= \frac{1}{1-\beta}(T_\tau^{k+1}\tau\xi_{k+1} - \beta T_\tau^k \tau\xi_k) + \epsilon_{k+1} - \gamma P_{k+1}(\epsilon_k + \epsilon_{\Delta_k}), \qquad (70)$$

$$= \frac{1}{1-\beta}(T_\tau^\star \tau\xi_{k+1} - \beta T_\tau^\star \tau\xi_k) + \epsilon_{k+1} - \gamma P_{k+1}(\epsilon_k + \epsilon_{\Delta_k}), \qquad (71)$$

where for $(i)$ we used Lemma 4.3 and the definition of $q_k$. Now using this inequality in the definition of $\xi_{k+1}$ gives

$$\tau\xi_{k+1} = (1-\beta) \sum_{i=0}^k \beta^{k-i} q_i, \qquad (72)$$

$$\geq (1-\beta)\beta^k q_0 + (1-\beta) \sum_{i=1}^k \beta^{k-i} \left( \frac{1}{1-\beta}(T_\tau^\star \tau\xi_i - \beta T_\tau^\star \tau\xi_{i-1}) + \epsilon_i - \gamma P_i(\epsilon_{i-1} + \epsilon_{\Delta_{i-1}}) \right), \qquad (73)$$

$$= T_\tau^\star \tau\xi_k - \beta^k T_\tau^\star \xi_0 + (1-\beta) \sum_{i=0}^k \beta^{k-i}(\epsilon_i - \epsilon_i') + (1-\beta)\beta^k Q_\tau^0, \qquad (74)$$

with $\epsilon_0' = 0$ and $\forall i > 0 : \epsilon_i' = \gamma P_i(I + \gamma P(I - \gamma P_i)^{-1}(\pi_{i-1} - \pi_i))\epsilon_{i-1}$. Letting $E_j := (1-\beta)\sum_{i=0}^j \beta^{j-i}(\epsilon_i - \epsilon_i')$, $R_m := R_x + \gamma\tau \log|A|$ be an upper bound to $\|T_\tau^\star \xi_0\|_\infty$ and $\bar{R} = \frac{R_m}{1-\gamma}$

be an upper bound to $\left\|Q_\tau^0\right\|_\infty$, we finally obtain

$$Q_\tau^\star - \tau\xi_{k+1} \le Q_\tau^\star - T_\tau^\star\tau\xi_k + \beta^k T_\tau^\star\xi_0 - E_k + (1-\beta)\beta^k Q_\tau^0, \tag{75}$$

$$\Rightarrow \left\|Q_\tau^\star - \tau\xi_{k+1}\right\|_\infty \le \left\|Q_\tau^\star - T_\tau^\star\tau\xi_k\right\|_\infty + \beta^k\left\|T_\tau^\star\xi_0\right\|_\infty + \left\|E_k\right\|_\infty + (1-\beta)\beta^k\left\|Q_\tau^0\right\|_\infty, \tag{76}$$

$$\le \gamma\left\|Q_\tau^\star - \tau\xi_k\right\|_\infty + \beta^k R_m + (1-\beta)\beta^k\bar{R} + \left\|E_k\right\|_\infty, \tag{77}$$

$$\le \gamma^{k+1}\left\|Q_\tau^\star\right\|_\infty + (R_m + (1-\beta)\bar{R})\sum_{i=0}^k \gamma^i\beta^{k-i} + \sum_{i=0}^k \gamma^i\left\|E_{k-i}\right\|_\infty, \tag{78}$$

$$= \gamma^{k+1}\left\|Q_\tau^\star\right\|_\infty + \frac{2-\gamma-\beta}{1-\gamma}R_m\sum_{i=0}^k \gamma^i\beta^{k-i} + \sum_{i=0}^k \gamma^i\left\|E_{k-i}\right\|_\infty \tag{79}$$

$\square$

## A.6 PROOF OF EQ. (14)

*Proof.* For $k=0$,

$$\xi_1 = \beta \times 0 + \alpha q_0 + \frac{\alpha\beta^M}{1-\beta^M}(q_0 - 0), \tag{80}$$

$$= \alpha\left(1 + \frac{\beta^M}{1-\beta^M}\right)q_0, \tag{81}$$

$$= \frac{\alpha}{1-\beta^M}q_0. \tag{82}$$

If it is true for $k$, then

$$\xi_{k+1} = \beta\frac{\alpha}{1-\beta^M}\sum_{i=0}^{M-1}\beta^i q_{k-1-i} + \alpha q_k + \frac{\alpha\beta^M}{1-\beta^M}(q_k - q_{k-M}), \tag{83}$$

$$= \frac{\alpha}{1-\beta^M}\sum_{i=0}^{M-2}\beta^{i+1} q_{k-1-i} + \frac{\alpha\beta^M}{1-\beta^M}(q_{k-M} - q_{k-M}) + \frac{\alpha}{1-\beta^M}q_k, \tag{84}$$

$$= \frac{\alpha}{1-\beta^M}\sum_{i=0}^{M-1}\beta^i q_{k-i} \tag{85}$$

$\square$

## A.7 PROOF OF THEOREM 5.1

As with policy iteration, we first need a policy improvement lemma

**Lemma A.1** (Approximate policy improvement of finite memory EPMD). *For any $k \ge 0$, $Q_\tau^{k+1} \ge Q_\tau^k - \gamma(I - \gamma P\pi_{k+1})^{-1}P[(\pi_{k+1} - \pi_k)(\epsilon_k + \Delta_k)]$, with $\Delta_k := \frac{\beta^M}{1-\beta^M}(q_k - q_{k-M})$.*

*Proof.* We can see the policy $\pi_{k+1}$ as the maximizer of Eq. (9) if we would replace $q_k$ with $q_k + \frac{\beta^M}{1-\beta^M}(q_k - q_{k-M})$. From the non-negativity of the $D_{\mathrm{KL}}$ and the fact that $D_{\mathrm{KL}}(\pi_k; \pi_k) = 0$ we have

$$\pi_k q_k - \tau h(\pi_k) \le \pi_{k+1}q_k - \tau h(\pi_{k+1}) + \frac{\beta^M}{1-\beta^M}(\pi_{k+1} - \pi_k)(q_k - q_{k-M}), \tag{86}$$

$$\Rightarrow \pi_k Q_\tau^k - \tau h(\pi_k) \le \pi_{k+1}Q_\tau^k - \tau h(\pi_{k+1}) + \frac{\beta^M}{1-\beta^M}(\pi_{k+1} - \pi_k)(q_k - q_{k-M}) + (\pi_{k+1} - \pi_k)\epsilon_k. \tag{87}$$

Let $\Delta_k := \frac{\beta^M}{1-\beta^M}(q_k - q_{k-M})$. Writing down the (matrix) definition of $Q_\tau^{k+1}$ and $Q_\tau^k$ gives

$$Q_\tau^{k+1} - Q_\tau^k = \gamma P(\pi_{k+1}Q_\tau^{k+1} - \tau h(\pi_{k+1})) - \gamma P(\pi_k Q_\tau^k - \tau h(\pi_k)), \tag{88}$$

$$\overset{(i)}{\geq} \gamma P(\pi_{k+1}Q_\tau^{k+1} - \tau h(\pi_{k+1})) - \gamma P(\pi_{k+1}Q_\tau^k - \tau h(\pi_{k+1}) + (\pi_{k+1} - \pi_k)(\epsilon_k + \Delta_k)), \tag{89}$$

$$= \gamma P\pi_{k+1}(Q_\tau^{k+1} - Q_\tau^k) - \gamma P[(\pi_{k+1} - \pi_k)(\epsilon_k + \Delta_k)], \tag{90}$$

$$\Leftrightarrow Q_\tau^{k+1} - Q_\tau^k \overset{(ii)}{\geq} -\gamma(I - \gamma P\pi_{k+1})^{-1}P[(\pi_{k+1} - \pi_k)(\epsilon_k + \Delta_k)]. \tag{91}$$

where in $(i)$ we have used the fact that $P$ is a probability matrix with only positive entries, and similarly in $(ii)$ for the matrix $(I - \gamma P\pi_{k+1})^{-1} = \sum_{i=0}^{\infty}(\gamma P\pi_{k+1})^i$. $\qquad\square$

We are now ready to prove the main theorem

*Proof.* The beginning of the proof is similar to vanilla entropy regularized policy/value iteration

$$\pi_{k+1}q_k - \tau h(\pi_{k+1}) = \pi_{k+1}\left(\frac{1}{1-\beta}(\tau\xi_{k+1} - \beta\tau\xi_k)\right) - \tau h(\pi_{k+1}) - \pi_{k+1}\Delta_k, \tag{92}$$

$$= \pi_{k+1}\left(\frac{1}{1-\beta}[\tau\xi_{k+1} - \tau h(\pi_{k+1}) - \beta(\tau\xi_k - \tau h(\pi_{k+1}))]\right) - \pi_{k+1}\Delta_k, \tag{93}$$

$$\overset{(i)}{\geq} \frac{1}{1-\beta}[\pi_{k+1}\tau\xi_{k+1} - \tau h(\pi_{k+1}) - \beta(\pi_k\tau\xi_k - \tau h(\pi_k))] - \pi_{k+1}\Delta_k. \tag{94}$$

Using this inequality in $q_{k+1}$ yields

$$q_{k+1} = Q_\tau^{k+1} + \epsilon_{k+1}, \tag{95}$$

$$= R + \gamma P(\pi_{k+1}Q_\tau^{k+1} - \tau h(\pi_{k+1})) + \epsilon_{k+1}, \tag{96}$$

$$\overset{(i)}{\geq} R + \gamma P(\pi_{k+1}(q_k - \epsilon_k - \gamma\mu_{k+1}P(\pi_{k+1} - \pi_k)(\epsilon_k + \Delta_k)) - \tau h(\pi_{k+1})) + \epsilon_{k+1}, \tag{97}$$

$$\tag{98}$$

where for $(i)$ we used Lemma 4.3 and the definition of $q_k$. Using Eq. (94) on the following terms gives

$$R + \gamma P(\pi_{k+1}q_k - \tau h(\pi_{k+1})) \geq R + \gamma P\left(\frac{1}{1-\beta}[\pi_{k+1}\tau\xi_{k+1} - \tau h(\pi_{k+1}) - \beta(\pi_k\tau\xi_k - \tau h(\pi_k))] - \pi_{k+1}\Delta_k\right), \tag{99}$$

$$= \frac{1}{1-\beta}(T_\tau^{k+1}\tau\xi_{k+1} - \beta T_\tau^k\tau\xi_k) - \gamma P\pi_{k+1}\Delta_k, \tag{100}$$

$$= \frac{1}{1-\beta}(T_\tau^\star\tau\xi_{k+1} - \beta T_\tau^\star\tau\xi_k) - \gamma P\pi_{k+1}\Delta_k. \tag{101}$$

Completing with the rest of the terms finally gives

$$q_{k+1} \geq \frac{1}{1-\beta}(T_\tau^\star\tau\xi_{k+1} - \beta T_\tau^\star\tau\xi_k) - \gamma P\pi_{k+1}(I + \gamma\mu_{k+1}P(\pi_{k+1} - \pi_k))(\epsilon_k + \Delta_k) + \epsilon_{k+1}. \tag{102}$$

$$\tag{103}$$

Let $A_{k+1} = \gamma P\pi_{k+1}(I + \gamma\mu_{k+1}P(\pi_{k+1} - \pi_k))$ using this inequality in the definition of $\xi_{k+1}$ gives

$$\tau\xi_{k+1} = \frac{1-\beta}{1-\beta^M} \sum_{i=0}^{M-1} \beta^i q_{k-i}, \tag{104}$$

$$\geq \frac{1-\beta}{1-\beta^M} \sum_{i=0}^{M-1} \beta^i \left( \frac{1}{1-\beta}(T_\tau^\star \tau\xi_{k-i} - \beta T_\tau^\star \tau\xi_{k-i-1}) - A_{k-i}(\epsilon_{k-i-1} + \Delta_{k-i-1}) + \epsilon_{k-i} \right), \tag{105}$$

$$= \frac{1}{1-\beta^M} T_\tau^\star \tau\xi_k - \frac{\beta^M}{1-\beta^M} T_\tau^\star \xi_{k-M} + \frac{1-\beta}{1-\beta^M} \sum_{i=0}^{M-1} \beta^i(\epsilon_{k-i} - A_{k-i}(\epsilon_{k-i-1} + \Delta_{k-i-1})). \tag{106}$$

Letting $E_j := \frac{1-\beta}{1-\beta^M} \sum_{i=0}^{M-1} \beta^i(\epsilon_{k-i} - A_{k-i}\epsilon_{k-i-1})$, $\Delta_q^k = \frac{1-\beta}{1-\beta^M} \sum_{i=0}^{M-1} \beta^i A_{k-i}\Delta_{k-i-1}$, $R_m := R_x + \gamma\tau \log|A|$ be an upper bound to $\|T_\tau^\star \xi_0\|_\infty$ and $\bar{R} = \frac{R_m}{1-\gamma}$ be an upper bound to $\|Q_\tau^0\|_\infty$, we finally obtain

$$Q_\tau^\star - \tau\xi_{k+1} \leq Q_\tau^\star - \frac{1}{1-\beta^M} T_\tau^\star \tau\xi_k + \frac{\beta^M}{1-\beta^M} T_\tau^\star \xi_{k-M} - E_k - \Delta_q^k, \tag{107}$$

$$\Rightarrow \|Q_\tau^\star - \tau\xi_{k+1}\|_\infty \leq \frac{\|Q_\tau^\star - T_\tau^\star \tau\xi_k\|_\infty + \beta^M \|Q_\tau^\star - T_\tau^\star \tau\xi_{k-M}\|_\infty}{1-\beta^M} + \|\Delta_q^k\|_\infty + \|E_k\|_\infty, \tag{108}$$

$$\leq \gamma\frac{\|Q_\tau^\star - \tau\xi_k\|_\infty + \beta^M \|Q_\tau^\star - \tau\xi_{k-M}\|_\infty}{1-\beta^M} + \|\Delta_q^k\|_\infty + \|E_k\|_\infty \tag{109}$$

Let us now look into the term $\|\Delta_q^k\|_\infty$, and split it into policy evaluation error and distance to $Q_\tau^\star$

$$\|\Delta_q^k\|_\infty = \left\| \frac{1-\beta}{1-\beta^M} \sum_{i=0}^{M-1} \beta^i A_{k-i}\Delta_{k-i-1} \right\|_\infty, \tag{110}$$

$$= \left\| \frac{1-\beta}{1-\beta^M} \sum_{i=0}^{M-1} \beta^i A_{k-i}\frac{\beta^M}{1-\beta^M}(q_{k-i-1} - q_{k-i-1-M}) \right\|_\infty, \tag{111}$$

$$= \left\| \frac{\beta^M(1-\beta)}{(1-\beta^M)^2} \sum_{i=0}^{M-1} \beta^i A_{k-i}(Q_{k-i-1} - Q_{k-i-1-M} + \epsilon_{k-i-1} - \epsilon_{k-i-1-M}) \right\|_\infty, \tag{112}$$

$$\leq \left\| \frac{\beta^M(1-\beta)}{(1-\beta^M)^2} \sum_{i=0}^{M-1} \beta^i A_{k-i}(\epsilon_{k-i-1} - \epsilon_{k-i-1-M}) \right\|_\infty$$
$$+ \left\| \frac{\beta^M(1-\beta)}{(1-\beta^M)^2} \sum_{i=0}^{M-1} \beta^i A_{k-i}(Q_{k-i-1} - Q_{k-i-1-M}) \right\|_\infty. \tag{113}$$

To bound the infinite norm of $A_k$, we note that $(1-\gamma)\mu_k$ is a probability matrix (the state distribution induced by policy $\pi_k$). Using the sub-aditivity of norms and the fact that the multiplication of probability matrices is a probability matrix with infinite norm equal to 1, we have

$$\|A_k\|_\infty = \left\| \gamma P\pi_{k+1}\left( I + \frac{\gamma}{1-\gamma}((1-\gamma)\mu_{k+1})P(\pi_{k+1} - \pi_k) \right) \right\|_\infty, \tag{114}$$

$$\leq \gamma + \frac{2\gamma^2}{1-\gamma}. \tag{115}$$

Looking now at the rightmost inner sum in Eq. (113) gives

$$\left\| \sum_{i=0}^{M-1} \beta^i A_{k-i}(Q_{k-i-1} - Q_{k-i-1-M}) \right\|_\infty \leq \sum_{i=0}^{M-1} \beta^i \left\| A_{k-i}(Q_{k-i-1} - Q_{k-i-1-M}) \right\|_\infty, \quad (116)$$

$$\leq \left( \gamma + \frac{2\gamma^2}{1-\gamma} \right) \sum_{i=0}^{M-1} \beta^i \left\| Q_{k-i-1} - Q_{k-i-1-M} \right\|_\infty, \quad (117)$$

$$\leq \left( \gamma + \frac{2\gamma^2}{1-\gamma} \right) \sum_{i=0}^{M-1} \beta^i \left( \left\| Q_\tau^\star - Q_{k-i-1} \right\|_\infty + \left\| Q_\tau^\star - Q_{k-i-1-M} \right\|_\infty \right), \quad (118)$$

$$\overset{(i)}{\leq} \frac{2}{1-\gamma} \left( \gamma + \frac{2\gamma^2}{1-\gamma} \right) \sum_{i=0}^{M-1} \beta^i \left( \left\| Q_\tau^\star - \tau\xi_{k-i-1} \right\|_\infty + \left\| Q_\tau^\star - \tau\xi_{k-i-1-M} \right\|_\infty \right), \quad (119)$$

where $(i)$ is due to Lemma 4.1. Let $z_k = \left\| Q_\tau^\star - \tau\xi_k \right\|_\infty$ and $T_k$ grouping all the error terms

$$T_k = \left\| E_k \right\|_\infty + \frac{\beta^M(1-\beta)}{(1-\beta^M)^2} \left\| \sum_{i=0}^{M-1} \beta^i A_{k-i}(\epsilon_{k-i-1} - \epsilon_{k-i-1-M}) \right\|_\infty. \quad (120)$$

Putting everything together we have

$$z_{k+1} \leq \gamma \frac{z_k + \beta^M z_{k-M}}{1-\beta^M} + \frac{\beta^M(1-\beta)}{(1-\beta^M)^2} \frac{2(\gamma+\gamma^2)}{(1-\gamma)^2} \sum_{i=0}^{M-1} \beta^i (z_{k-i-1} + z_{k-i-1-M}) + T_k. \quad (121)$$

Let us first study the sequence $\{z_k\}$ without policy evaluation errors and try to upper bound it with a simpler sequence. We define the sequence $\{x_k\}$ for all integers $k$ by

$$x_k = \left\| Q_\tau^\star \right\|_\infty, \text{ for all } k \leq 0, \quad (122)$$

and for $k \geq 0$ we let

$$x_{k+1} = \gamma \frac{x_k + \beta^M x_{k-M}}{1-\beta^M} + \frac{\beta^M(1-\beta)}{(1-\beta^M)^2} \frac{2(\gamma+\gamma^2)}{(1-\gamma)^2} \sum_{i=0}^{M-1} \beta^i (x_{k-i-1} + x_{k-i-1-M}). \quad (123)$$

We first find a condition for which the sequence is strictly decreasing starting from $k \geq 0$. For $k = 0$ we have that

$$x_1 = \left( \gamma \frac{1+\beta^M}{1-\beta^M} + \frac{\beta^M}{1-\beta^M} \frac{4(\gamma+\gamma^2)}{(1-\gamma)^2} \right) x_0. \quad (124)$$

This will be strictly decreasing if

$$\gamma \frac{1+\beta^M}{1-\beta^M} + \frac{\beta^M}{1-\beta^M} \frac{4(\gamma+\gamma^2)}{(1-\gamma)^2} < 1, \quad (125)$$

$$\Leftrightarrow \gamma(1+\beta^M) + \beta^M \frac{4(\gamma+\gamma^2)}{(1-\gamma)^2} < (1-\beta^M), \quad (126)$$

$$\Leftrightarrow \left( 1 + \gamma + \frac{4(\gamma+\gamma^2)}{(1-\gamma)^2} \right) \beta^M < 1-\gamma, \quad (127)$$

$$\Leftrightarrow \log \left( 1 + \gamma + \frac{4(\gamma+\gamma^2)}{(1-\gamma)^2} \right) + M \log \beta < \log(1-\gamma), \quad (128)$$

$$\Leftrightarrow M > \log \frac{(1-\gamma)^3}{(1+\gamma)(1-\gamma)^2 + 4(\gamma+\gamma^2)} / \log \beta. \quad (129)$$

As it is true for $k = 0$ and the sequence is constant for $k < 0$, assume now that the sequence is strictly decreasing from there on, up to some positive index $k$. Then since all the weights of past terms are positive, we can replace all terms by their predecessors and we immediately have that

$$x_{k+1} < \gamma \frac{x_{k-1} + \beta^M x_{k-M-1}}{1 - \beta^M} + \frac{\beta^M (1 - \beta)}{(1 - \beta^M)^2} \frac{2(\gamma + \gamma^2)}{(1 - \gamma)^2} \sum_{i=0}^{M-1} \beta^i (x_{k-i-2} + x_{k-i-2-M}), \quad (130)$$

$$= x_k. \quad (131)$$

Thus the sequence $\{x_k\}$ is non-increasing for all $k$ if $M$ satisfies the inequality in (129). For such values of $M$ we will now find an upper bounding sequence that has a simpler geometric form. Letting $c = \frac{\beta^M}{1 - \beta^M} \left( \frac{4(\gamma + \gamma^2)}{(1 - \gamma)^2} + \gamma \right)$, and since the sequence is non-decreasing, we have for all $k \geq 0$ that

$$x_{k+1} \leq \frac{\gamma}{1 - \beta^M} x_k + c x_{k-2M}. \quad (132)$$

Let us now try to find a rate $d \in (0, 1)$ such that for all $k$ we have

$$x_k \leq d^k x_0. \quad (133)$$

For all $k \leq 0$, the above inequality holds for any $d \in (0, 1)$. Now, if the upper bounding is true up to some index $k$ then using Eq. (132) we have

$$x_{k+1} \leq \left( \frac{\gamma}{1 - \beta^M} d^k + c d^{k-2M} \right) x_0, \quad (134)$$

$$= d^k \left( \frac{\gamma}{1 - \beta^M} + c d^{-2M} \right) x_0. \quad (135)$$

The smallest acceptable $d$ would be one such that

$$\frac{\gamma}{1 - \beta^M} + c d^{-2M} = d, \quad (136)$$

$$\Leftrightarrow d^{2M+1} - \frac{\gamma}{1 - \beta^M} d^{2M} - c = 0. \quad (137)$$

Let $f(d) = d^{2M+1} - \frac{\gamma}{1 - \beta^M} d^{2M} - c$, we have that $f(\frac{\gamma}{1 - \beta^M}) = -c < 0$ and that $f(1) = 1 - \frac{\gamma}{1 - \beta^M} - c > 0$ from the above condition on $M$. Since $f$ is continuous and increasing between these two values it accepts a unique root $d_0 \in (\frac{\gamma}{1 - \beta^M}, 1)$ which would satisfy the sought geometric sequence upper bound for all $k$.

We now turn to the part of the sequence of $z_k$ that depends on the error terms $T_k$. Define for $k \leq 0$

$$y_k = 0, \quad (138)$$

and for $k \geq 0$

$$y_{k+1} = \gamma \frac{y_k + \beta^M y_{k-M}}{1 - \beta^M} + \frac{\beta^M (1 - \beta)}{(1 - \beta^M)^2} \frac{2(\gamma + \gamma^2)}{(1 - \gamma)^2} \sum_{i=0}^{M-1} \beta^i (y_{k-i-1} + y_{k-i-1-M}) + T_k. \quad (139)$$

Let $\bar{T} = \max_{0 \leq i \leq k} T_i$, and $\gamma_M = \frac{\gamma}{1 - \beta^M}$. Then we have that

$$y_k \leq \frac{\bar{T}}{1 - (\gamma_M + c)}. \quad (140)$$

Indeed, it is true for $k \leq 0$, and if it is true up to $k$ then

$$y_{k+1} \leq (\gamma_M + c) \frac{\bar{T}}{1 - (\gamma_M + c)} + T_k, \quad (141)$$

$$\leq (\gamma_M + c) \frac{\bar{T}}{1 - (\gamma_M + c)} + \bar{T}, \quad (142)$$

$$= \frac{\bar{T}}{1 - (\gamma_M + c)}. \quad (143)$$

Replacing in Eq. (139), we have

$$y_{k+1} \leq \gamma_M y_k + c \frac{\bar{T}}{1 - (\gamma_M + c)} + T_k, \tag{144}$$

$$\leq \gamma_M^{k+1} y_0 + \sum_{i=0}^{k} \gamma_M^i \left( T_{k-i} + c \frac{\bar{T}}{1 - \gamma_M - c} \right), \tag{145}$$

$$= \sum_{i=0}^{k} \gamma_M^i \left( T_{k-i} + c \frac{\bar{T}}{1 - \gamma_M - c} \right). \tag{146}$$

Finally, because the upper bound $z_{k+1}$ in Eq. (121) has linear dependencies on previous $z_i$ terms ($i \leq k$), we immediately have that $z_k \leq x_k + y_k$. Indeed, it is true for $k = 0$, since $z_0 = \|Q_\tau^\star\|_\infty = x_0 + y_0$. And if we assume that it is true for $k$, using Eq. (121), we immediatly have that it is true for $k + 1$. Thus

$$z_{k+1} \leq x_{k+1} + y_{k+1}, \tag{147}$$

$$\leq d_0^{k+1} \|Q_\tau^\star\|_\infty + \sum_{i=0}^{k} \gamma_M^i \left( T_{k-i} + c \frac{\bar{T}}{1 - \gamma_M - c} \right). \tag{148}$$

$\square$

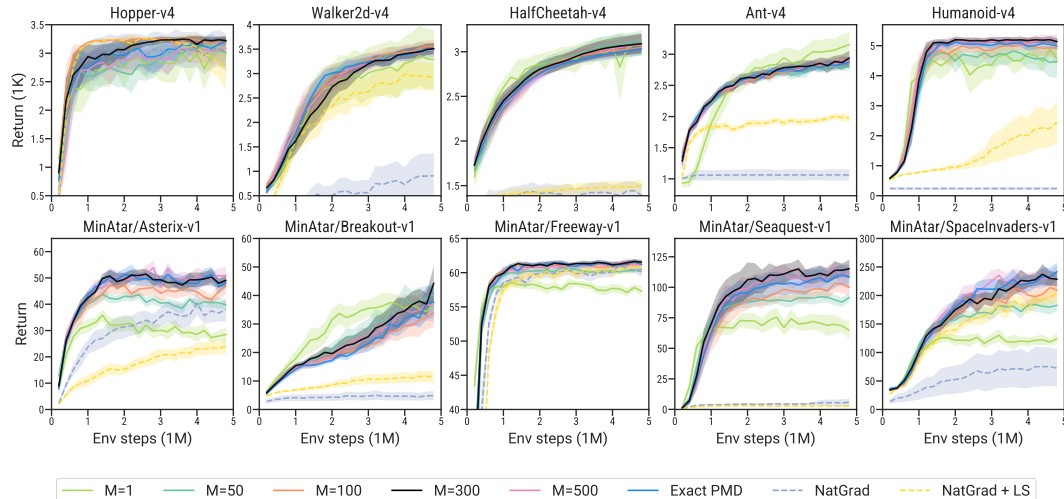

Figure 3: Evaluation of StaQ for different memory sizes $M$, on MuJoCo and MinAtar environments. Results show mean and 95% confidence interval for 20 seeds.

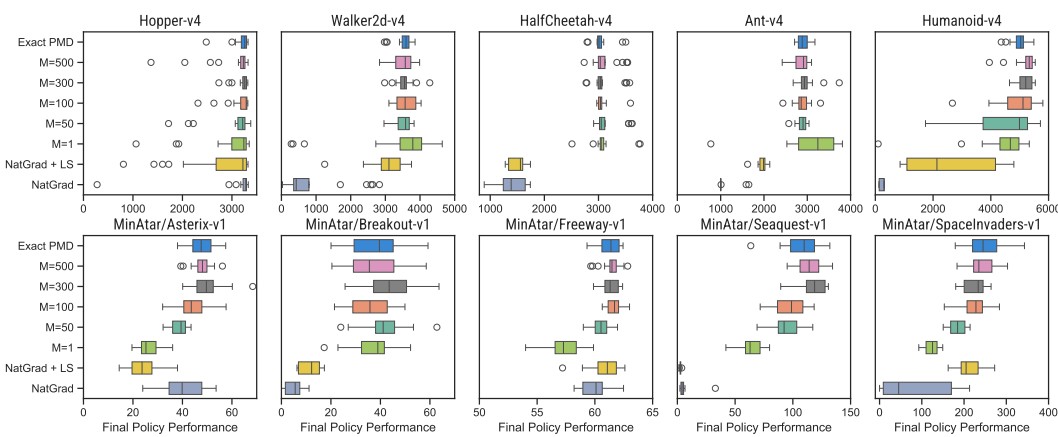

Figure 4: Box plots of final policy performance of StaQ for different memory sizes $M$, on MuJoCo and MinAtar environments for 20 seeds.

## B  EXPERIMENTAL RESULTS

### B.1  THE IMPACT OF THE MEMORY-SIZE M

Figure 3 shows the mean and 95% confidence interval of the performance of StaQ for different choices of $M$ on MuJoCo and MinAtar tasks. Figure 4 shows the distribution of the final policy performance across seeds for each algorithm and task. These results are averaged over 20 seeds. To provide higher statistical confidence of our results, for Humanoid and Acrobot, we show the mean and 95% confidence intervals in Table 2 and 3, evaluated over 30 seeds. Setting $M = 1$ corresponds to no KL-regularization as discussed in App. C.1 and can be seen as an adaptation of SAC to discrete action spaces. $M = 1$ is unstable on both Hopper and Walker. Adding KL-regularization and averaging over at least 50 Q-functions greatly helps to stabilize performance except on the Humanoid task and some MinAtar tasks, where $M = 50$ was still unstable compared to exact EPMD. In general we note diminishing returns in increasing $M$, with $M = 100$ matching the performance of exact EPMD in most environment and $M = 300$ being virtually indistinguishable from exact EPMD. Compared to natural policy gradient, with and without line search, performance is generally improved across all tasks, making StaQ a potentially better alternative to natural policy gradient, at least on environments where Q-function computations can be efficiently batched.

| | M=1 | M=50 | M=100 | M=300 | M=500 | Exact EPMD |
|---|---|---|---|---|---|---|
| Humanoid-v4 | $4272 \pm 282$ | $4600 \pm 367$ | $4795 \pm 215$ | $5143 \pm 90$ | $5102 \pm 132$ | $5166 \pm 108$ |

Table 2: Performance on Humanoid for various values of $M$. Mean and 95% confidence interval, across 30 seeds.

| | M=1 | M=5 | M=50 | M=100 | M=300 | M=500 |
|---|---|---|---|---|---|---|
| Acrobot-v4 | $-62.69 \pm 0.48$ | $-62.69 \pm 0.29$ | $-62.44 \pm 0.13$ | $-62.49 \pm 0.12$ | $-62.54 \pm 0.17$ | $-62.34 \pm 0.08$ |

Table 3: Performance on Acrobot for various values of $M$. Mean and 95% confidence interval, across 30 seeds.

### B.2 HYPERPARAMETER OF NATGRAD + LS

The baseline "NatGrad + LS" introduces an additional hyperparameter $\epsilon_{\mathrm{KL}}$ compared to the standard PMD setting that StaQ follows. To properly set this hyperparameter we perform a preliminary study using 3 seeds and a subset of the MuJoCo and MinAtar tasks. The results in Figure 5 show that on a few tasks, the value of the $\epsilon_{\mathrm{KL}}$ does not impact greatly the performance, but when there are significant differences between the two algorithms, $\epsilon_{\mathrm{KL}} = 0.001$ is often better. As such, we use $\epsilon_{\mathrm{KL}} = 0.001$ for all experiments involving "NatGrad + LS".

### B.3 COMPARISON WITH DEEP RL BASELINES

We summarize all performance comparisons with the deep RL baselines in Fig. 6 and Table 4. We provide a discussion of the MountainCar environment and some of the challenges of exploration in an entropy-regularized setting in App. B.5.

| | StaQ (M=300) | M-DQN | DQN | PPO | TRPO |
|---|---|---|---|---|---|
| CartPole-v1 | **500** | 457 | 411 | **500** | **500** |
| Acrobot-v1 | **-62** | -63 | -63 | -63 | -64 |
| LunarLander-v2 | **285** | 88 | -317 | 227 | 222 |
| MountainCar-v0 | -200 | **-100** | -110 | -141 | -118 |
| Hopper-v4 | **3196** | 2600 | 2279 | 2411 | 2672 |
| Walker2d-v4 | **3550** | 1364 | 1424 | 2799 | 3010 |
| HalfCheetah-v4 | **3061** | 2098 | 2294 | 2001 | 1731 |
| Ant-v4 | **2910** | 1776 | 1871 | 2277 | 2452 |
| Humanoid-v4 | **5273** | 2580 | 2887 | 588 | 700 |
| MinAtar/Asterix-v1 | **46** | 31 | 19 | 9 | 23 |
| MinAtar/Breakout-v1 | 48 | **55** | 34 | 10 | 15 |
| MinAtar/Freeway-v1 | **62** | 59 | 54 | 60 | 47 |
| MinAtar/Seaquest-v1 | **114** | 51 | 14 | 5 | 7 |
| MinAtar/SpaceInvaders-v1 | **242** | 116 | 95 | 92 | 94 |

Table 4: Final performance on all environments.

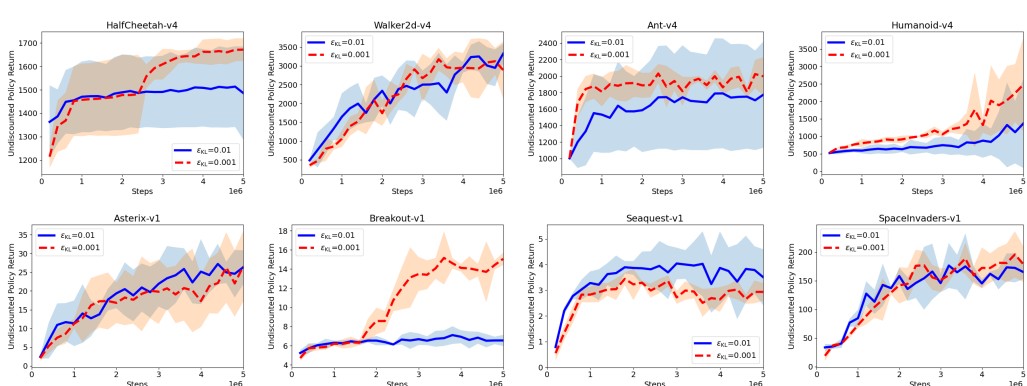

Figure 5: Mean and standard deviation of policy performance of "NatGrad + LS" baseline for two values of $\epsilon_{\text{KL}}$ averaged over 3 seeds.

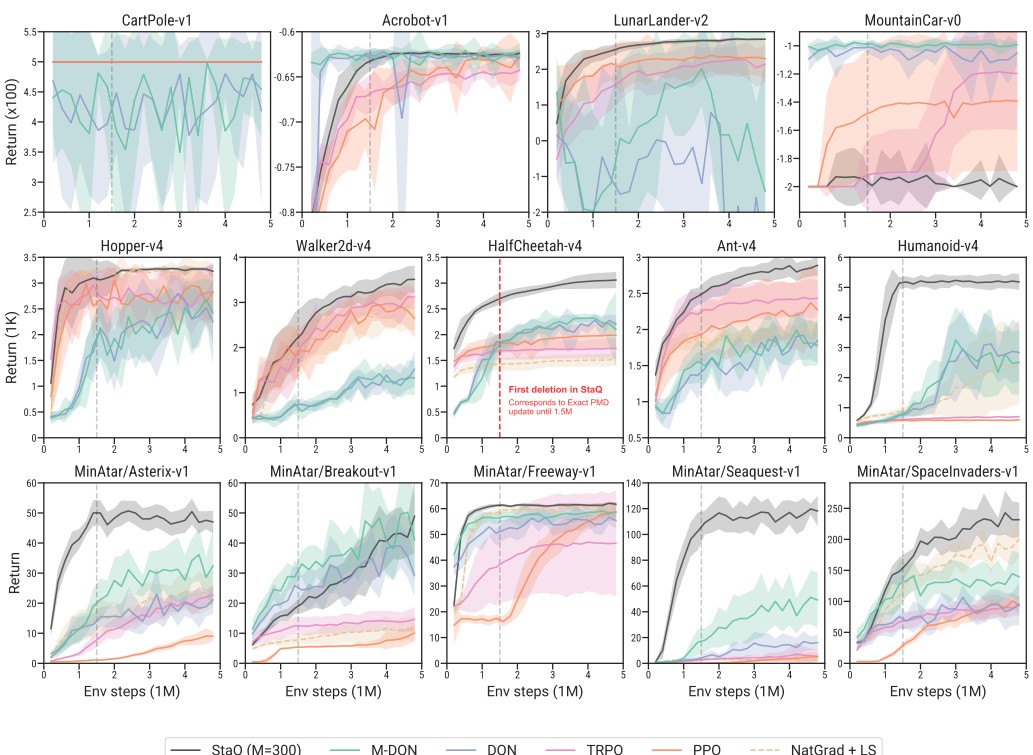

Figure 6: Policy performance of StaQ ($M = 300$) vs deep RL baselines across all environments. Results showing the mean and standard deviation across 10 seeds.

### B.4 STABILITY PLOTS (VARIATION WITHIN INDIVIDUAL RUNS)

In this section we provide plots that demonstrate the intra seed oscilliations. In Fig. 7-8 we plot the returns of the first three seeds of the full results (shown in Fig. 6). At each timestep, the returns for each individual seed are normalised by subtracting and then dividing by the mean across all seeds. In addition to the first three seeds, the shaded regions indicate one-sided tolerance intervals such that at least 95% of the population measurements are bounded by the upper or lower limit, with confidence level 95% (Krishnamoorthy & Mathew, 2009).

We can see from Fig. 7-8 that Approximate Policy Iteration (API) algorithms (StaQ with M=300, TRPO, PPO) generally exhibit less variation within runs than Approximate Value Iteration (AVI) ones (DQN, M-DQN). In simple environments, such as CartPole, all three API algorithms have stable performance, but on higher dimensional tasks, only StaQ retains a similar level of stability while maintaining good performance. This is especially striking on Hopper, where runs show comparatively little variation within iterations while having the highest average performance, as shown in Fig. 6. We attribute this improved stability in the performance of the evaluation policy by the averaging over a very large number of Q-functions ($M = 300$) of StaQ, which reduces the infamous performance oscillation of deep RL algorithms in many cases.

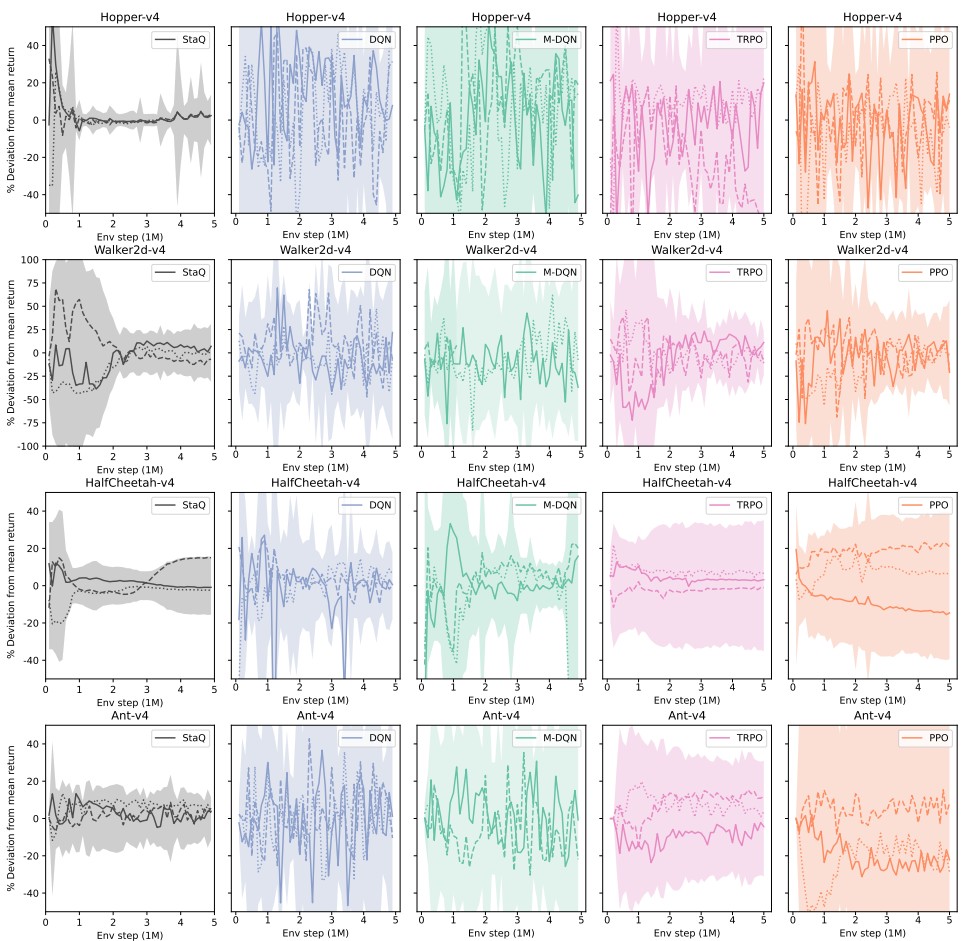

Figure 7: Stability plots for MuJoCo environments, plotting normalized performance of the first three individual runs for each algorithm. See text for more details. Figures continue on the next page.

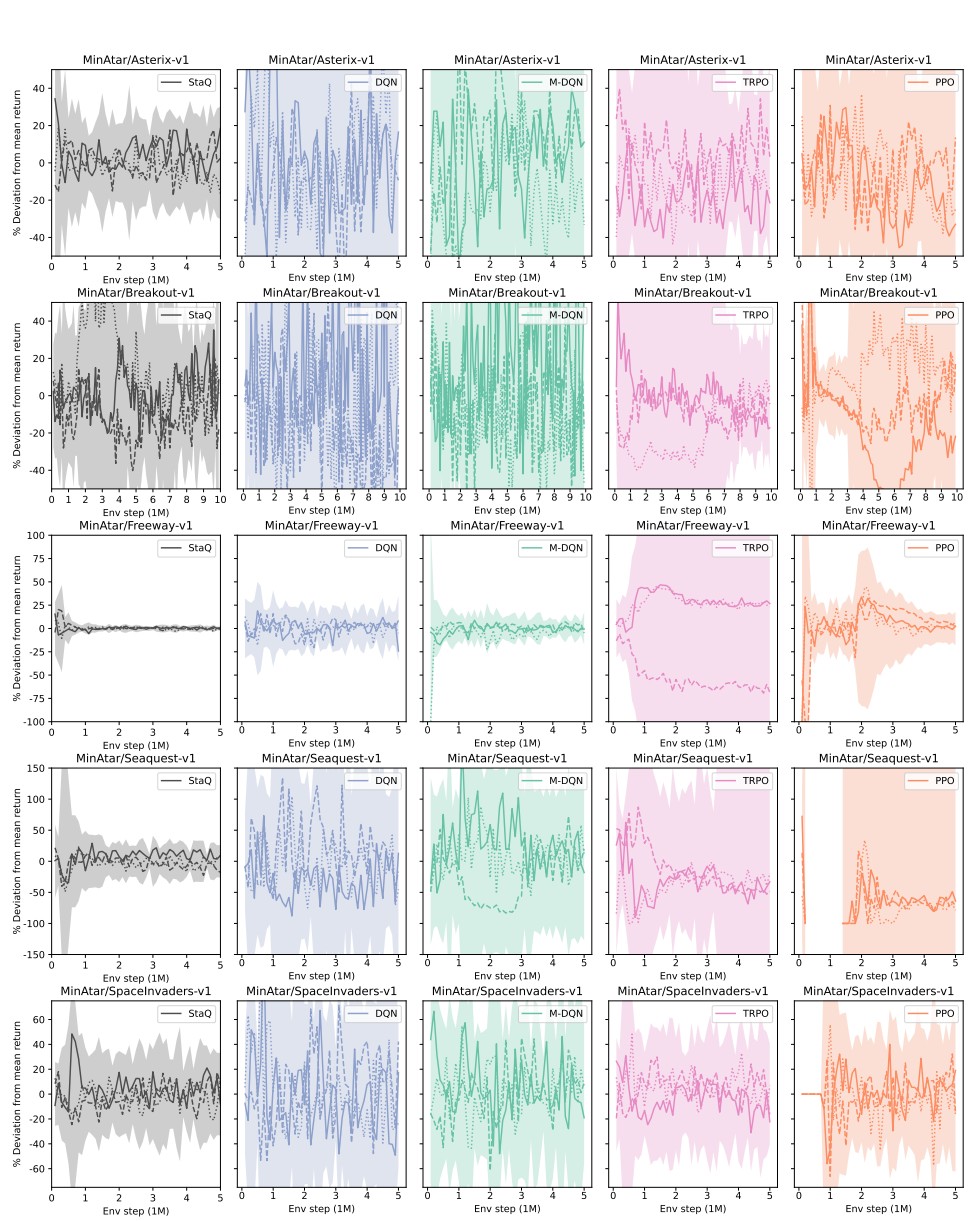

Figure 8: Stability plots for MinAtar environments, plotting normalized performance of the first three individual runs for each algorithm. See text for more details.

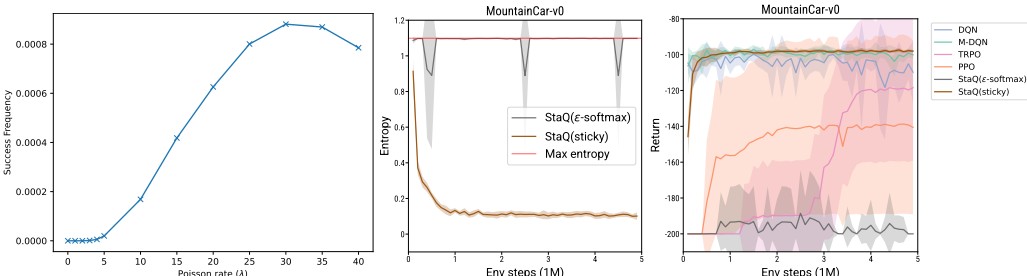

Figure 9: **Left:** Frequency of non-zero rewards of a uniform policy with sticky actions for different choice of Poisson rate $\lambda$ on MountainCar over 5M timesteps. **Middle:** Entropy of learned policies under different behavior policies. Entropy of the uniform (Max entropy) policy plotted for reference. **Right:** Policy returns for StaQ with different behavior policies and deep RL baselines on Mountain-Car. Adding sticky actions to StaQ's behavior policy fixes its performance on this task.

### B.5 ENTROPY REGULARIZATION DOES NOT SOLVE EXPLORATION

StaQ and exact EPMD achieve strong performance on all 14 environments except on MountainCar where they fail to learn. In this section, we perform additional experiments to understand the failure of StaQ on MountainCar. In short, it appears that the initial uniform policy—which has maximum entropy—acts as a strong (local) attractor for this task: StaQ starts close to the uniform policy, and exploration with this policy does not generate a reward signal in MountainCar. As StaQ does not observe a reward signal in early training, it quickly converges to the uniform policy which has maximum entropy, but also never generates a reward signal. Indeed, if we unroll a pure uniform policy on MountainCar for 5M steps, we will never observe a reward.

However, StaQ is not limited to a specific choice of behavior policy, and choosing a policy that introduces more correlation between adjacent actions, like a simple "sticky" policy allows StaQ to solve MountainCar. This policy samples an action from $\pi_k$ and applies it for a few consecutive steps, where a number of steps is drawn randomly from Poisson($\lambda$) distribution (in our experiments with StaQ we fix the rate of Poisson distribution at $\lambda = 10$). In Fig. 9, we can see that StaQ with the same hyperparameters for classical environments (see Table 6) and a "sticky" behavior policy manages to find a good policy for MountainCar matching the best baseline. The final policy demonstrates much lower entropy compared to the default policy that fails at learning for this environment. We focused on this paper on the benefits of entropy regularization in averaging evaluation error. While it is believed that entropy might help with exploration, these observations are a good reminder that entropy regularization remains a heuristic exploration strategy that does not replace a theoretically grounded strategy, which is beyond the scope of this paper that only focuses on using entropy regularization to reduce the error floor of approximate policy iteration when using function approximators.

## C ADDITIONAL ACTOR-CRITIC BASELINES

### C.1 RELATION BETWEEN STAQ AND SOFT ACTOR-CRITIC

In this section, we explain the relation between Soft Actor-Critic (SAC, Haarnoja et al. (2018)) and both M-DQN (Vieillard et al., 2020b) and StaQ with $M = 1$. SAC is not directly used as a baseline because SAC is not compatible with discrete action spaces. However, M-DQN can be seen as an adaptation of SAC to discrete action spaces with an additional KL-divergence regularizer. Please see the discussion in Vieillard et al. (2020b) on page 3, between Eq. (1) and (2). Vieillard et al. (2020b) also describe Soft-DQN in Eq. (1) as a straightforward discrete-action version of SAC, that can be obtained from M-DQN by simply setting the KL-divergence regularization weight to zero. Soft-DQN was not included as a baseline because the results of Vieillard et al. (2020b) suggest that M-DQN generally outperforms Soft-DQN. Our results already suggest that adding a $D_{\mathrm{KL}}$ regularizer generally improves performance, and the results in the next section further confirm this.

We also note that by setting $M = 1$ in StaQ, we remove the KL-divergence regularization and only keep the entropy bonus. This baseline can also be seen as an adaptation of SAC to discrete action spaces: indeed, if we set $M = 1$ in Eq. (2) we recover the policy logits

$$\xi_{k+1} = \frac{\alpha}{1 - \beta^M} \sum_{i=0}^{M-1} \beta^i Q_\tau^{k-i} \tag{149}$$

$$= \frac{\alpha}{1 - \beta} Q_\tau^k \tag{150}$$

$$= \frac{Q_\tau^k}{\tau}, \tag{151}$$

where the last line is due to $\alpha\tau = 1 - \beta$. This results in a policy of the form $\pi_{k+1} \propto \exp\left(\frac{Q_\tau^k}{\tau}\right)$. Meanwhile, for SAC, the actor network is obtained by minimizing the following problem (Eq. 14 in Haarnoja et al. (2018)) for states sampled from some replay buffer $\mathcal{D}$

$$\pi_{k+1} = \arg\min_\pi \mathbb{E}_{s \sim \mathcal{D}} \left[ \mathrm{KL}\left( \pi(s) \middle\| \frac{\exp\left(\frac{Q_\tau^k(s)}{\tau}\right)}{Z_{\mathrm{norm.}}} \right) \right] \tag{152}$$

However, in the discrete action setting, we can sample directly from $\exp\left(\frac{Q_\tau^k}{\tau}\right)$—which is the minimizer of the above KL-divergence term—and we do not need an explicit actor network. As such StaQ with $M = 1$ could be seen as an adaptation of SAC to discrete action spaces. However, we discuss next a method with an explicit actor that is even closer to SAC.

## C.2    ACTOR-CRITIC POLICY MIRROR DESCENT METHODS

We introduce now several additional actor-critic baselines that can be derived from the entropy regularized PMD update in Eq. 9. As discussed in the previous section, in the absence of a $D_{\mathrm{KL}}$ regularizer, the solution of the policy update problem is $\exp\left(\frac{Q_\tau^k}{\tau}\right)$, which corresponds to StaQ with $M = 1$ in our experiments. While we can sample directly from this policy, we consider also an actor-critic version thereof that updates the actor by attempting to solve Eq. 152 via gradient descent, resulting in an even closer baseline to SAC. We label this baseline AC-NoKL.

Next, if we add $D_{\mathrm{KL}}$ regularization to the policy update, the entropy regularized PMD solution becomes $\pi_{k+1} \propto \pi_k^\beta \exp\left(\alpha Q_\tau^k\right)$. Tracking this recursive policy definition exactly results in an infinite sum, but we can approximate $\pi_k^\beta \exp\left(\alpha Q_\tau^k\right)$ with an actor network. Specifically, we consider a second baseline inspired by ECPO (Mei et al., 2019) that advocates for minimizing the M-Projection between the actor and the PMD solution

$$\pi_{k+1} = \arg\min_\pi \mathbb{E}_{s \sim \mathcal{D}} \left[ \mathrm{KL}\left( \frac{\pi_k^\beta \exp\left(\alpha Q_\tau^k\right)}{Z_{\mathrm{norm.}}} \middle\| \pi(s) \right) \right], \tag{153}$$

where states are again sampled from a replay buffer $\mathcal{D}$. The authors of ECPO argue that the M-Projection will better preserve the support of the distribution and prevent the premature elimination of actions during exploration. We will label this baseline AC-MProj.

Finally, we consider an actor-critic baseline inspired by MDPO (Tomar et al., 2022) that optimizes a state averaged version of the entropy regularized PMD update, i.e.

$$\pi_{k+1} = \arg\max_\pi \mathbb{E}_{s \sim \mathcal{D}} \left[ Q_\tau^k(s) \cdot \pi(s) - \tau h(\pi(s)) - \eta D_{\mathrm{KL}}(\pi(s); \pi_k(s)) \right], \tag{154}$$

by performing a few gradient steps over the above objective. Unlike ECPO, MDPO skips the closed form solution and optimizes directly the entropy and $D_{\mathrm{KL}}$ regularized objective by gradient ascent. We will call this baseline AC-DirectOpt. An important parameter for MDPO and ECPO is $m$, the number of gradient steps that are used to update the actor. Before performing the full scale evaluation of AC-DirectOpt and AC-MProj—the two AC methods following the updates of MDPO and ECPO respectively, we first perform a 3 seeds hyper-parameter sensitivity analysis w.r.t. $m$ on 4 MinAtar tasks. Results show in Fig. 10 and Fig. 11 that on some tasks, this parameter is

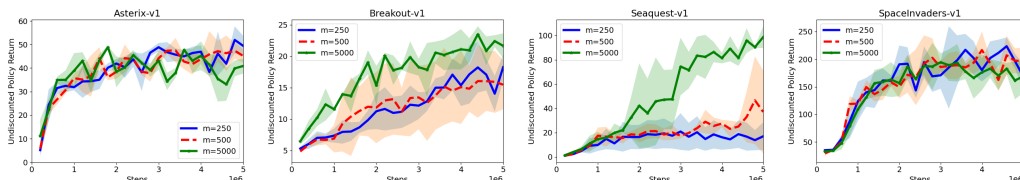

Figure 10: Sensitivity of AC-DirectOpt to the number of gradient steps performed to update the actors. Plots show mean and std. deviation over 3 seeds. See App. C.2 for details.

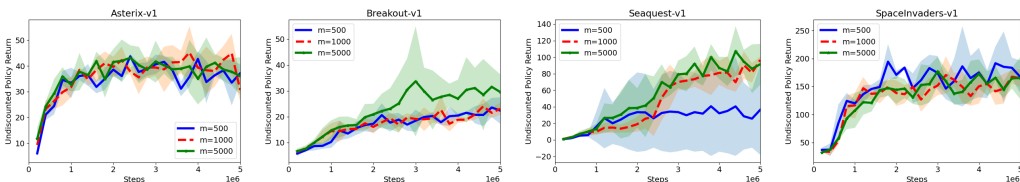

Figure 11: Sensitivity of AC-MProj to the number of gradient steps performed to update the actors. Plots show mean and std. deviation over 3 seeds. See App. C.2 for details.

actually not very crucial, but on some other tasks, using more gradient steps for the actor seems to improve performance. As such, for the remained of this section, we will use $m = 5000$ for both AC-DirectOpt and AC-MProj, which corresponds to performing one actor gradient update per time-step in the environment. This is the same setting as in SAC, and we will use the same number of actor updates for the AC-NoKL baseline.

We now perform a 20 seeds comparisons on all 5 MinAtar tasks between StaQ and the three AC baselines discussed above, isolating again as in the main experiment the policy update from the rest of the RL components that are kept identical. Fig. 12 shows the performance of the current policy evaluated every 200K time-steps using 25 rollouts, Fig. 13 shows the distribution of final policy performance, and Fig. 14 shows aggregate results on the 5 tasks. As can be seen, AC-DirectOpt and AC-MProj being approximations of Exact PMD, end-up having worse performance on several tasks. Comparing the AC methods between themselves, AC-DirectOpt and AC-MProj have close performance on the aggregate metrics, while AC-NoKL lags behind the other two and shows again the benefits of $D_{\mathrm{KL}}$ regularization on policy update. Compared to StaQ, the performance of the $D_{\mathrm{KL}}$ regularized AC methods (i.e. AC-DirectOpt and AC-MProj) is between that of StaQ with $M = 1$ and $M = 50$, while AC-NoKL performs worse than $M = 1$. The improved performance of StaQ with higher $M$ comes at little performance cost as can be seen in Table 5, where the run time remains close to that of $M = 1$, whereas all three AC methods have to perform double the amount of gradient steps to update the actor in addition to the critic, resulting in a 35 to 45% longer run time than StaQ with $M = 1$.

# D HYPERPARAMETERS

Here, we provide the full list of hyperparameters used in our experiments[2]. StaQ's hyperparameters are listed in Table 6, while the hyperparameters for our baselines are provided in Tables 7-9. For TRPO and PPO, we use the implementation provided in `stable-baselines4` (Raffin et al., 2021), while we used our in-house PyTorch implementation of (M)-DQN[3].

To account for the different scales of the reward between environments, we apply a different reward scaling to the Classic/MuJoCo environments and MinAtar. Note that this is equivalent to inverse-scaling the entropy weight $\tau$ and KL weight $\eta$, ensuring that $\xi_k$ is of the same order of magnitude for all environments. To account for the varying action dimension $|A|$ of the environments, we set the *scaled entropy coefficient* $\bar{\tau}$ as a hyperparameter, defined by $\bar{\tau} = \tau \log |A|$, rather than directly

---

[2]Code is provided in the supplemental zip file, and will be released to an open-source repository upon publication.

[3]We will add the link to the in-house library upon publication.

Table 5: Runtime of StaQ and AC PMD baselines on MinAtar/SapceInvaders-v1 on an NVIDIA Tesla V100 GPU.

| | M=1 | M=100 | M=300 | AC-NoKL | AC-MProj | AC-DirectOpt |
|---|---|---|---|---|---|---|
| MinAtar/SpaceInvaders-v1 | 9.6 | 10.4 | 10.5 | 13.9 | 13.1 | 13.2 |

Figure 12: Policy evaluations during training of StaQ with different values of $M$ and actor-critic PMD baselines. Mean and 95% confidence interval over 20 seeds.

setting $\tau$. Furthermore, the entropy weight is linearly annealed from its minimum and maximum values.

**Policy evaluation.** In all our experiments, we use an ensemble of two neural networks, similarly to e.g. SAC (Haarnoja et al., 2018), to evaluate a $Q$-function and therefore two SNNs for $\xi$-logits. In particular, we optimize the current Q-function weights $\theta$ to minimize the loss $\mathcal{L}(\theta)$,

$$\mathcal{L}(\theta) = \mathbb{E}_{(s,a)\sim\mathcal{D}} \left[ \frac{1}{2} \left( Q_\theta\left(s,a\right) - \hat{Q}\left(s,a\right) \right)^2 \right] \tag{155}$$

$$\hat{Q}(s,a) := R(s,a) + \gamma\mathbb{E}_{s'\sim\mathcal{D},a\sim\pi(s')} \left[ \text{agg}_{i\in\{1,2\}} Q_{\hat{\theta}_i}(s',a') - \tau h(\pi(s')) \right] \tag{156}$$

where $\text{agg}$ computes either the $\texttt{min}$ or $\texttt{mean}$ over the target Q-functions with weights $\hat{\theta}_1, \hat{\theta}_2$. We find that using the $\texttt{min}$ of the two Q-functions to compute the target values often results in more stable training. $\texttt{min}$ gives a more conservative target that is robust to overestimation bias in the Q-functions, and this allows us to reduce the KL weight. However, such a strategy may struggle when reward is not dense enough, e.g. some MinAtar and some classic control environments. Therefore we instead use the $\texttt{mean}$ in Classic/MinAtar environments. Future work could use a more sophisticated approach that is both robust to overestimation bias and yet sensitive to weak reward signals.

### D.1 IMPACT OF $\epsilon$ EXPLORATION

In our main experiment we use $\epsilon = 0.05$ as the probability to sample an action uniformly at random. In a set of additional experiments we evaluate the impact $\epsilon$ has on performance. To this end, we launch StaQ with $M = 300$ and Exact PMD that never deletes a Q-function within the 5 million time-step window for different values of $\epsilon$ on the 5 MinAtar tasks. We see that $\epsilon$ exploration is important for both Breakout and Seaquest, but has less of an impact on other tasks. In fact, on Asterix and SpaceInvaders, increasing the epsilon decreases the performance slightly, perhaps because of the increase to the off-policiness of the data which might deteriorate policy evaluation. As such, our choice of $\epsilon = 0.05$ in the main experiment appears to provide a good trade-off between improving exploration and limiting the off-policiness of data on MinAtar tasks. In all cases, independent of the choice of $\epsilon$, StaQ with $M = 300$ remains close to Exact PMD in performance.

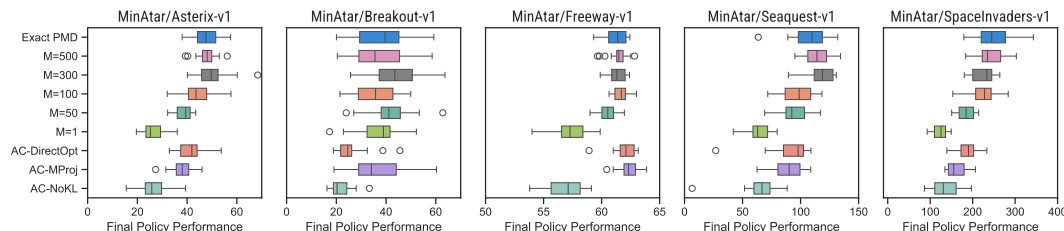

Figure 13: Box plot for final policy performance of StaQ with different values of $M$ and actor-critic PMD baselines, over 20 seeds.

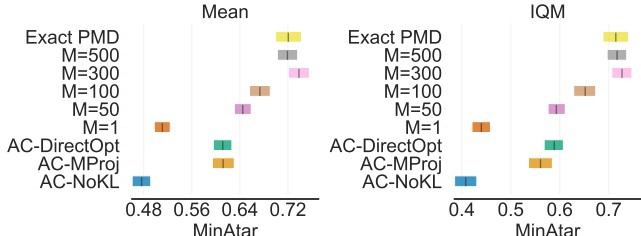

Figure 14: Aggregate final performance for StaQ with different values of $M$ and actor-critic PMD baselines, for MinAtar tasks over 20 seeds.

| Hyperparameter | Classic | MuJoCo | MinAtar |
|---|---|---|---|
| Discount ($\gamma$) | 0.99 | 0.99 | 0.99 |
| Memory size ($M$) | 300 | 300 | 300 |
| Policy update interval | 5000 | 5000 | 5000 |
| Ensembling mode | **mean** | **min** | **mean** |
| Target type | hard | hard | hard |
| Target update interval | 200 | 200 | 200 |
| Epsilon | 0.05 | 0.05 | 0.05 |
| Reward scale | **10** | **10**[*] | **100** |
| KL weight ($\eta$) | **20** | **10** | **20** |
| Initial scaled ent. weight | 2.0 | 2.0 | 2.0 |
| Final scaled ent. weight | 0.4 | 0.4 | 0.4 |
| Ent. weight decay steps | **500K** | **1M** | **1M** |
| Architecture | $256 \times 2$ | $256 \times 2$ | Conv(16, 3, 3) + 128 MLP |
| Activation function | ReLU | ReLU | ReLU |
| Learning rate | 0.0001 | 0.0001 | 0.0001 |
| Optimizer | Adam | Adam | Adam |
| Replay capacity | 50K | 50K | 50K |
| Batch size | 256 | 256 | 256 |

Table 6: StaQ hyperparameters, with parameters which vary across environment types in bold. [*]Hopper-V4 uses a reward scale of 1.

| Hyperparameter | Classic | MuJoCo | MinAtar |
|---|---|---|---|
| Discount factor ($\gamma$) | 0.99 | 0.99 | 0.99 |
| Horizon | 2048 | 2048 | 1024 |
| Num. epochs | 10 | 10 | 3 |
| Learning starts | 5000 | 20000 | 20000 |
| GAE parameter | 0.95 | 0.95 | 0.95 |
| VF coefficient | 0.5 | 0.5 | 1 |
| Entropy coefficient | 0 | 0 | 0.01 |
| Clipping parameter | 0.2 | 0.2 | $0.1 \times \alpha$ |
| Optimizer | Adam | Adam | Adam |
| Architecture | $64 \times 2$ | $64 \times 2$ * | Conv(16, 3, 3) + 128 MLP |
| Activation function | Tanh | Tanh | Tanh |
| Learning rate | $3 \times 10^{-4}$ | $3 \times 10^{-4}$ | $2.5 \times 10^{-4} \times \alpha$ |
| Batch size | 64 | 64 | 256 |

Table 7: PPO hyperparameters, based on (Schulman et al., 2017). In the MinAtar environments $\alpha$ is linearly annealed from 1 to 0 over the course of learning. *Humanoid-v4 uses a hidden layer size of 256.

| Hyperparameter | Classic | MuJoCo | MinAtar |
|---|---|---|---|
| Discount factor ($\gamma$) | 0.99 | 0.99 | 0.99 |
| Horizon | 2048 | 2048 | 2048 |
| Learning starts | 5000 | 20000 | 20000 |
| GAE parameter | 0.95 | 0.95 | 0.95 |
| Stepsize | 0.01 | 0.01 | 0.01 |
| Optimizer | Adam | Adam | Adam |
| Architecture | $64 \times 2$ | $64 \times 2$ * | Conv(16, 3, 3) + 128 MLP |
| Activation function | Tanh | Tanh | Tanh |
| Learning rate | $3 \times 10^{-4}$ | $3 \times 10^{-4}$ | $2.5 \times 10^{-4}$ |
| Batch size | 64 | 64 | 256 |

Table 8: TRPO hyperparameters, based on (Schulman et al., 2015). *Humanoid-v4 uses a hidden layer size of 256.

| Hyperparameter | Classic | MuJoCo | MinAtar |
|---|---|---|---|
| Discount factor ($\gamma$) | 0.99 | 0.99 | 0.99 |
| Target update interval | 100 | 8000 | 8000 |
| Epsilon | 0.1 | 0.1 | 0.1 |
| Decay steps | 20K | 20K | 20K |
| M-DQN temperature | 0.03 | 0.03 | 0.03 |
| M-DQN scaling term | 1.0 | 0.9 | 0.9 |
| M-DQN clipping value | -1 | -1 | -1 |
| Architecture | $512 \times 2$ | $128 \times 2$ | Conv(16, 3, 3) + 128 MLP |
| Activation function | ReLU | ReLU | ReLU |
| Learning rate | $1 \times 10^{-3}$ | $5 \times 10^{-5}$ | $2.5 \times 10^{-4}$ |
| Optimizer | Adam | Adam | Adam |
| Replay capacity | 50K | 1M | 1M |
| Batch size | 128 | 32 | 32 |

Table 9: MDQN and DQN hyperparameters, based on (Vieillard et al., 2020b; Ceron & Castro, 2021)

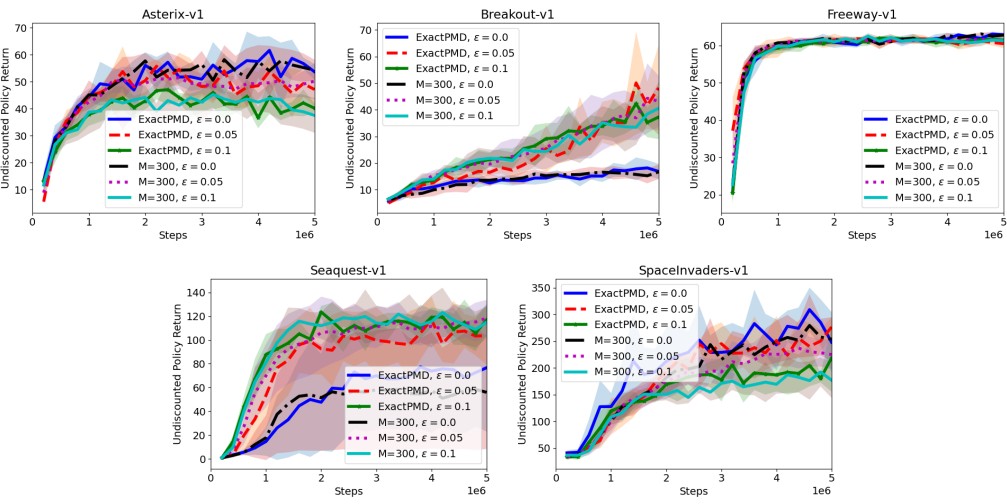

Figure 15: Impact of the $\epsilon$ exploration parameter of StaQ on performance in MinAtar tasks. Plots showing mean and std. deviation over 5 seeds.

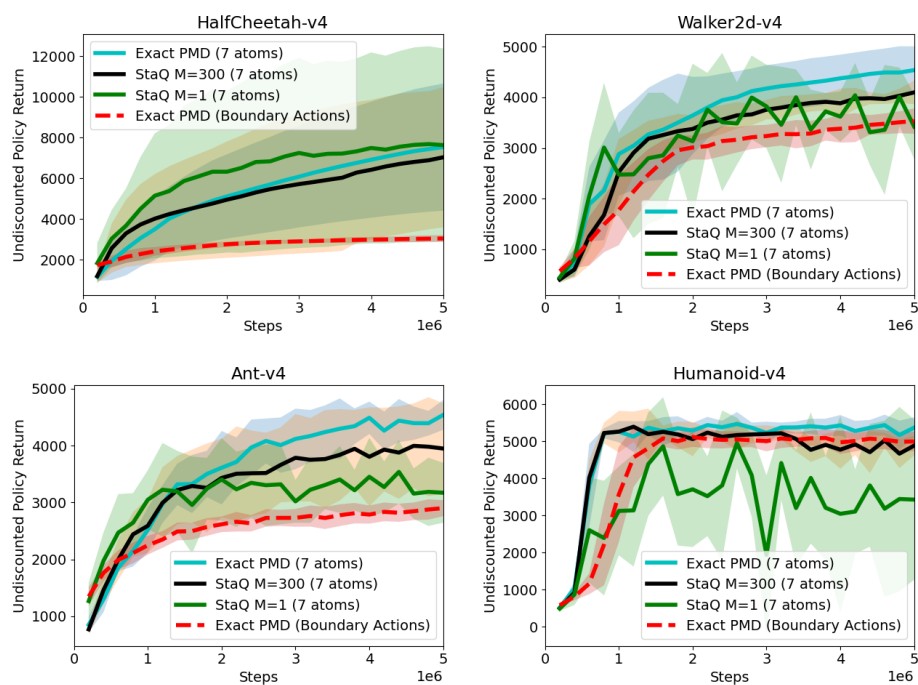

Figure 16: StaQ on MuJoCo with action space discretized using the independence assumption from Tang & Agrawal (2020) with 7 atoms. Plots show improvements over the discretization that only considers boundary actions on all but the Humanoid-v4 task. Results averaged over 5 seeds.

# E EXTENSION TO CONTINUOUS ACTION SPACES

The main challenge of StaQ in continuous action spaces is to sample from its Boltzmann policy with logits given by Eq. 14. Straightforward extensions of StaQ to continuous action spaces can be to distill the policy into an easier to sample from actor (e.g. Gaussian neural policy), or to use the Gumbel-max trick to sample from StaQ's policy via optimization. While applying the Gumbel-max trick might be slow, for StaQ's theory to hold, we only need accurate samples when computing the targets for the current Q-function (and these sampled actions can be pre-computed as discussed in Sec. 5.1 for the logits). For data gathering however, the behavior policy could use less accurate samples from StaQ's policy to speed-up inference. Nonetheless, we expect this solution as well as distilling the StaQ into a more compact actor to yield additional policy update errors compared to exact PMD and StaQ. As discussed in Sec. 4.3, Zhan et al. (2023) showed that PMD with approximate policy updates have a higher error floor (see Thm. 2 in Zhan et al.'s paper and the dependence to $\epsilon_{opt}$) and might converge to lower quality solutions. As such, these extensions do not fit claims of the paper, which is that for some environments and thanks to the continuous advances in GPU hardware, it can become more advantageous to stack Q-functions to get an error free policy update with low computational overhead, instead of following the more traditional actor-critic approach to PMD.

To tackle continuous actions in the spirit of the contributions of our paper, one could use StaQ instead as the gating policy in a hierarchical RL framework. For instance, starting from our MuJoCo experiments, one could consider the hand discretized actions we used as sub-policies and further optimize them via gradient ascent. The sub-policies could be more complex, for instance PID controllers or neural based as in the work of Gehring et al. (2021); Hafner et al. (2022), that use a discrete set of sub-policies and would thus preserve the above properties of fast and accurate policy updates for the gating policy if optimized by truncated PMD.

Yet another direction to tackle continus action spaces is to leverage an independence assumption in the policy following Tang & Agrawal (2020). Concretely, Tang & Agrawal discretize each action dimension into a finite number of atoms and assume that each action dimension follows an inde-

pendent distribution—which is also assumed by continuous action RL algorithms using diagonal Gaussian policies. In our setting, this translate in the logit space, which is also the space of Q-functions, by assuming separability of the Q-function approximation i.e. that $q_k$, the approximation of the Q-function $Q_k$, is given by

$$q_k(s,a) = \sum_{i=1}^{\dim(A)} q_k^{[i]}(s, a^{[i]}), \tag{157}$$

where $a^{[i]}$ is entry $i$ of the action vector $a$ and $\{q_k^{[i]}\}_{i=1}^{\dim(A)}$ is a set of real functions. This independence assumption makes it possible to consider fine grained discretizations with extremely large but finite action spaces, from which one can sample exactly and efficiently by sampling each action dimension independently. Incidentally, this separability assumption could allow a more efficient implementation of the Gumbel-max trick discussed above.

To illustrate how this could benefit StaQ, we have implemented the discretization from Tang & Agrawal (2020) in combination with StaQ using $K = 7$ atoms by action dimension. Concretely, the Q-network outputs K values for each action dimension, and the Q-function for a given action is computed using Eq. 157. We have run 5 seeds of StaQ using this Q-network architecture on the 4 MuJoCo environments shown in Fig. 16. We have kept all hyper-parameters of StaQ similar except for the target update rate of the Q-target which we have increased to 1000 for HalfCheetah and Walker and to 2500 for Ant and Humanoid. From the plot, we can see large improvements due to the finer grained discretization on all but the Humanoid-v4 task. Moreover, using StaQ improves over previous results reported by Tang & Agrawal (2020), that used PPO and TRPO as base learners on most MuJoCo tasks, showing the impact our work could have in this line of research.

In summary, beyond environments that have naturally discrete action spaces or low dimensional continuous action spaces that can be easily discretized, we have outlined here several RL frameworks that tackle continuous actions spaces yet leverage Boltzmann policies and that could benefit from our work. We think interesting research questions in this direction is to investigate the improvements the Boltzmann distribution could bring to exploration compared to the unimodal Gaussian distribution that is commonly used in continuous action deep RL algorithms.

## F PSEUDOCODE OF STAQ

We provide in this section the pseudocode of StaQ in Alg. 1. As an approximate policy iteration algorithm, StaQ comprises three main steps: i) data collection, ii) policy evaluation iii) policy improvement. Data collection (Line 4-5) consist in interacting with the environment to collect transitions of type (state, action, reward, next state) that are stored in a replay buffer. A policy evaluation algorithm (Eq. 155) is then called to evaluate the current Q-function $Q_\tau^k$ using the replay buffer. Finally, the policy update is optimization-free and simply consists in stacking the Q-function in the SNN policy as discussed in Sec. 5.1. After $K$ iterations, the last policy is returned.

---

**Algorithm 1** StaQ (Finite-memory entropy regularized policy mirror descent)

---

1: **Input:** An MDP $\mathcal{M}$, a memory-size $M$, Number of samples per iteration $N$, Replay buffer size $D$, Initial behavior policy $\pi_0^b$, entropy weight $\tau$, $D_{\mathrm{KL}}$ weight $\eta$, $\epsilon$-softmax exploration parameter

2: **Output:** Policy $\pi_K \propto \exp(\xi_K)$
3: **for** $k = 0$ **to** $K - 1$ **do**
4:      Interact with $\mathcal{M}$ using the behavior policy $\pi_k^b$ for $N$ times steps
5:      Update replay buffer $\mathcal{D}_k$ to contain the last $D$ transitions
6:      Learn $Q_\tau^k$ from $\mathcal{D}_k$ using a policy evaluation algorithm (Eq. 155)
7:      Obtain logits $\xi_{k+1}$ by stacking the last $M$ Q-functions (see Sec. 5.1) following the finite-memory EPMD update of Eq. 14.
8:      Set $\pi_{k+1} \propto \exp(\xi_{k+1})$ and $\pi_{k+1}^b$ to an $\epsilon$-softmax policy over $\pi_{k+1}$
9: **end for**

---

