# OpenReview forum: "StaQ: a Finite Memory Approach to Discrete Action Policy Mirror Descent"
_ICLR.cc/2026/Conference — Submitted to ICLR 2026_

### Official Review · Reviewer_f4Z1 · 2025-10-24

**Soundness:** 1
**Presentation:** 1
**Contribution:** 1
**Rating:** 2
**Confidence:** 4

**Summary:**

The authors observe that the exact Policy Mirror Descent (PMD) algorithm produces a policy that depends on the sum of all past Q-functions. However, under function approximation, this accumulation introduces error over time. To address this issue, the authors restrict their analysis to the discrete action space and propose a PMD-like algorithm that retains the most recent M Q-functions. In this formulation, for a sufficiently large but finite M, the policy update error diminishes, while the averaging effect in policy evaluation remains (up to error terms that decay exponentially with respect to M). Furthermore, the authors propose a practical implementation of their algorithm to investigate the benefit of stacking past M Q-functions in practice.

**Strengths:**

From a theoretical standpoint, it is both important and interesting to explore whether incorporating the past M Q-functions can provide benefits in the function approximation setting. However, I am not convinced that the findings in the paper translate to the practical setting.

**Weaknesses:**

- The presentation of the paper is at times very difficult to follow.
- The paper is positioned as an investigation into whether the theoretical foundations of PMD translate effectively into practice. However, all empirical results are deferred to the appendix without a clear guideline on what to look for, which limits the reader’s ability to assess the validity of the hypotheses. After carefully reviewing the appendix, I have several concerns:
    - The theoretical analysis focuses on the discrete action setting, yet the experiments start with continuous control tasks where the policy is parameterized as a diagonal Gaussian distribution.
    - For MuJoCo environments, Soft Actor-Critic (SAC) is widely regarded as a strong baseline, but it is not included in the results. Conversely, DQN and M-DQN, which are designed for discrete action spaces, are evaluated on MuJoCo tasks.
    - In the Atari experiments, DQN and M-DQN use $\epsilon$-greedy exploration, but it is unclear whether the proposed method uses the same exploration strategy.
    - It is not specified whether the authors relied on a standard implementation framework (e.g., spinning up or stable-baselines) for the proposed method, which makes reproducibility and fair comparison difficult to assess.

**Questions:**

In the preliminaries, the policy is defined as  $\pi \propto \exp(Q(s, \cdot))$. However, based on the standard formulation of PMD, shouldn’t this instead be $\pi_t \propto \pi_{t-1} \exp(Q^{t-1}(s, \cdot))$? This would reflect the iterative update of the policy relative to the previous one, which, when applied recursively, results in the sum over all passed Q-functions?

In the function approximation setting, the early Q-functions are typically of low quality. Could you elaborate on the intuition behind why storing these past Q-functions would be beneficial? This design choice seems somewhat counterintuitive. Additionally, could you clarify which policy is used to sample actions during interaction with the environment?

---

> ### Author Response · Authors · 2025-11-22
> **Weaknesses**
>
> Dear reviewer, we thank you for all your feedback. We are glad that you find our theoretical contributions interesting. Below you will find our response to the weaknesses and questions you listed.
>
> **W1. Paper presentation difficult to follow and results in Appendix.**
> We have improved the experimental results in the main paper, adding more details of the experimental setup, as well as aggregated performance plots of StaQ for a range of memory sizes $M$. We hope that this makes the main paper more self-contained. For full details of the manuscript changes please see the global comment.
>
> If there remain specific aspects of the manuscript that are difficult to follow, please let us know and we will try to address these as best we can.
>
> **W2. Discrete action setting yet diagonal Gaussian policy.** We do not use a diagonal Gaussian policy, but use a finite number of actions that each apply one of the two extremum torque controls to each joint at a time, as discribed in line 447. Moreover, we have added Appendix E that includes a new set of experiments on MuJoCo following an alternative discretization that uses an independence of action dimension assumption (as in diagonal Gaussian policies in continuous action deep RL) based on the work of Tang et al. 2020. This assumption allows for finer grained discretization per dimension yet tractable learning and action sampling. The results shown in Fig. 10 shows that StaQ can also learn in this setting and it even appears to perform better on these control problems than PPO and TRPO that were used as the base RL learners by Tang et al. 2020, showing the potential impact StaQ could have on this line of research.
>
> *Reference:*
> * Tang et al.; Discretizing Continuous Action Space for On-Policy Optimization; AAAI, 2020
>
> **W3. SAC not included for MuJoCo despite being a strong baseline.** Default SAC implementations do not usually support discrete action spaces. However, $M=1$ uses soft Q-learning and no KL regularization and is a close replacement to SAC as discussed in Appendix C that relates SAC to M-DQN and StaQ with $M=1$. Indeed SAC performs well on MuJoCo and StaQ with $M=1$ seems to perform well too.
>
> **W4. DQN and M-DQN use $\epsilon$-greedy, not clear about StaQ.**
> Yes we add an $\epsilon$ probability of sampling a random action with $\epsilon=0.05$ on top of the current policy as stated in Line 454. We use the same exploration strategy for all tasks and experiments.
>
> **W5. Implementation framework not specified.**
> As described in the first paragraph of App D. (Line 1643), we use Stable-Baselines3 for PPO and TRPO, and an in-house library for DQN and M-DQN. Unfortunately, although this library is publicaly available, we are unable to share the details for anonymity reasons. We will share the full library + code for StaQ upon publication.

---

> > ### Author Response · Authors · 2025-11-22
> > **Questions**
> >
> > **Q1. PMD update formula.**
> > We apologize if this was confusing, but in Line 162, writing $\pi\propto\exp(Q)$ is only meant to introduce a notation that drops dependence on the state. We used the Q-function as an example, but this applies equally to the unnormalized logits functions $\xi:S\times A\mapsto \mathbb{R}$, which is also a real function of the state-action space. Throughout the paper, the current policy is $\pi_k\propto\exp(\xi_k)$ as stated in Eq. 8. The formula you provide is almost correct for vanilla entropy regularized PMD, only missing the temperatures. Taking the $\exp$ of Eq. 10, gives $\pi_{k+1} \propto \pi_k^\beta \exp(\alpha q_k)$. The recursive policy definiton of StaQ can be recovered from Eq. 14. We hope this clears out the confusion.
> >
> > **Q2. Intuition behind storing old Q-functions.**
> > The dependence on older Q-function is a direct consequence of the KL regularization. In very broad terms, this smoothens the greedification operator when computing the targets of Q-functions: without regularization, to approximate the next Q-function $Q_{k+1}$ we greedily pick the best action according to the previous Q estimate $q_k$. Since $q_k$ is only an approximation of $Q_k$, it might contain errors and we might pick the wrong action and jump back and forth between multiple wrong actions throughout the policy iteration  algorithm. Averaging over multiple Q-function estimates gives a chance for evaluation errors to cancel each other out and also stabilizes the distribution of maximizing action. This averaging effect directly shows in the different bounds of Thm 4.2, 4.4 and 5.1. We hope this helps somehow and are available to clarify further the intuition if needed.
> >
> > **Q2'. Early Q-functions are typically of low quality.**
> > We thank you for the excellent question. We agree that initial Q-functions due to the limited amount of gradient updates they receive might have higher evaluation errors. However, in StaQ and in most deep RL implementations, Q networks are initialized to very flat low magnitude functions over the state-action space. As such, since they are flat functions they do not influence the policy one way or another, since the policy is invariant to a constant shift to its logits. If initial Q-functions had a bigger impact on the algorithm, then we believe a finite $M$ would more clearly outperform the untruncated vanilla PMD, but this does not show in a significant way in our experiment. As such, we cannot state that this is a practical problem. However, if this becomes the case in some given setting, then the truncated PMD algorithm we propose could be a principled way to deal with it while preserving the averaging of error property as shown by Thm. 5.1.

---

> > > ### Comment · Reviewer_f4Z1 · 2025-11-25
> > >
> > > I thank the authors for their responses, but I remain concerned about several aspects of the experimental design. Methods such as M-DQN evaluate performance across 60 Atari games and demonstrate improvements over prior work such as IQN. In contrast, the authors restrict their evaluation to discrete MuJoCo tasks, where the discretization procedure itself introduces an additional confounding factor. The only genuinely discrete experiments are 5 MinAtar games, which are considerably simpler than Atari. In such settings, it is not surprising that storing past $Q$ functions does not degrade performance, as the state space is small enough that early $Q$ estimates are not severely harmful.
> > >
> > >
> > > For Atari, which is substantially more challenging, I expect that storing past $Q$ functions would negatively affect performance. Early $Q$ estimates are typically very inaccurate, and meaningful learning only emerges after significant training as the agent encounters more complex states. Moreover, for DQN, disabling $\epsilon$-greedy exploration can still yield reasonable performance, although using it provides a clear improvement. In contrast, I do not believe disabling $\epsilon$-greedy would be viable for the proposed method, as such exploration is precisely what would allow the agent to escape the poor $Q$ estimates accumulated from earlier snapshots.
> > >
> > > Overall, I remain unconvinced that storing past $Q$ functions is beneficial in practice, especially in more complex environments.

---

> > > > ### Author Response · Authors · 2025-11-26
> > > >
> > > > Thank you for your response. We would like to clarify the following points:
> > > >
> > > > **Storing old Q-functions.** Due to the exponential weighting of older Q-functions, the contribution of old Qs is vanishingly small. For example, for $\beta\simeq 0.95$, the weighting of Q-values more than 100 iterations old is $< 0.006$. This decay, a consequence of entropy regularisation, is one of the intuitive motivations for truncating the oldest Qs.
> > > >
> > > > Therefore, even for Exact PMD, with no truncation, the very old Q-functions will have a negligible contribution. This is even more true with truncation in place: for instance with M=100 in our MinAtar experiments, only the Q-functions learned in the last 500k timesteps out of the 5 million total are used.
> > > >
> > > >
> > > > **Exploration of DQN.** DQN has intrinsic exploration due to the phenomenon of policy churn (Schaul et al. 2022): after just a single update, the greedy policy of DQN changes in approximately 10% of all states. This is a significant secondary form of exploration present in DQN. Indeed, they show that if both $\epsilon$-greedy and policy churn are removed, the performance of DQN collapses (Fig. 2 of Schual et all 2022). The exploration of DQN is an accidental side-effect of the instability of neural networks.
> > > >
> > > > In contrast, the policy of StaQ exhibits much less policy churn, by design due to the KL-regularisation, but also as a consequence of the averaging over Q-functions (see e.g. the stability plots comparing StaQ and DQN in Figs 7-8, App. B.4). Due to this reduced policy churn, StaQ relies on explicit exploration mechanisms -- we chose $\epsilon$-greedy as a simple baseline.
> > > >
> > > >
> > > > **References:**
> > > >
> > > > - Schaul et al; The Phenomenon of Policy Churn; NeurIPS 2022.

---

> > > > > ### Comment · Reviewer_f4Z1 · 2025-11-27
> > > > >
> > > > > Thank you for the response. My concerns regarding the experimental design and the motivation for storing past $Q$ functions in non-toy environments remain unaddressed. In practical implementations, the parameters of the DQN target network are already maintained as an exponential moving average of the online network to stabilize training, and it is unclear how the proposed contribution would be useful in such practical settings (see the polyak_update function in stable-baselines implementation of DQN as an example). For these reasons, I will maintain my score.

---

> > > > > > ### Author Response · Authors · 2025-11-27
> > > > > >
> > > > > > Dear reviewer, averaging the weights of neural networks as in the Polyak update does not amount to averaging their outputs, otherwise yes the PMD formula would have been much easier :). The benefits of averaging Q-functions comes directly from the PMD formulation and its theoretical properties which are discussed extensively in the paper. And unlike many prior PMD studies, our algorithm is a one to one implementation of the algorithm analyzed theoretically. If you are aware of simpler methods to retain the desirable properties of PMD, or if you have references regarding the theoretical properties of Polyak averaging of target networks that are related to our work, we would be happy to discuss these in our paper. Thank you.

---

### Official Review · Reviewer_VUeC · 2025-10-30

**Soundness:** 3
**Presentation:** 3
**Contribution:** 3
**Rating:** 6
**Confidence:** 3

**Summary:**

StaQ is a finite-memory variant of Policy Mirror Descent for \emph{discrete} actions that averages only the most recent \(M\) Q-functions. A stacked-Q implementation keeps computation efficient. As \(M\) increases, StaQ closely matches exact PMD and yields more stable, higher returns on discretized MuJoCo and MinAtar.

**Strengths:**

1. The paper provides guarantees that the truncation error from keeping only $M$ past Q-functions rapidly diminishes as $M$  grows, so the policy-update bias effectively vanishes. This bridges the gap between an idealized infinite-history PMD and a practical algorithm, giving users a clear knob $M$ to trade memory for accuracy.

2. The stacked-Q architecture enables parallel evaluation of multiple historical Q estimates in a single forward pass, avoiding extra policy-optimization steps. In practice, this keeps wall-clock overhead modest even for large \(M\), which makes the method easy to scale on standard GPUs.

3. Across several discrete or discretized benchmarks, increasing \(M\) consistently improves return and reduces variability across random seeds. For sufficiently large \(M\) (e.g., hundreds), performance is nearly indistinguishable from exact PMD, supporting the theoretical claims with tangible gains.

**Weaknesses:**

1 .The method and analysis focus on  discrete actions; extensions to continuous control are not developed. Given that many modern RL tasks are continuous, this limits immediate applicability and may require nontrivial adaptations of both the algorithm and proofs.
2.  Several experiments rely on discretizing originally continuous-control environments, which can alter the dynamics and policy landscape. Conclusions drawn in these settings may not directly transfer to canonical continuous-control benchmarks without further validation.

**Questions:**

see weakness

---

> ### Author Response · Authors · 2025-11-21
> **Addressing weaknesses**
>
> Dear reviewer, we thank you for your feedback and we answer below your questions.
>
> **Q1. Extension of method and analysis to continuous actions.** In our proofs some elementary concepts such as transition matrices need to be generalized to the continuous case. However, we do not expect any hurdle in extending our analysis to continuous state-action spaces. We limit the scope of our paper to discrete action spaces rather for practical purposes, as our algorithm defines an implicit policy from the truncated some of past Q-functions which can only be sampled from efficiently in discrete action spaces. As suggested by Reviewer E3vT, extensions to continuous cases could be envisioned using the Gumbel-max trick, but we do not consider it in the current paper as this goes against the messaging of an error free and optimization free policy update leveraging batched GPU computations that we put forth in our paper. Instead, we believe more natural extensions of our work to continuous action spaces could be to use our algorithm for learning discrete gating policies in a hierarchical reinforcement learning setting, as discussed in the first reply to Reviewer E3vT and in the newly added Appendix E in the paper.
>
> Moreover, in this new appendix, we consider an alternative discretization approach following Tang et al. 2020. This approach leverages an independence assumption that allows for a finer grained discretization, bridging the gap between discrete and continuous action spaces. We show in a new set of experiments that StaQ also works using this finer grained discetization on MuJoCo and the results seem to improve over those of Tang et al. 2020 that used PPO and TRPO as base learners; showing the impact our work could have on this line of research. While such discretization could still lead to a loss of expressiveness, it raises interesting research questions on the improvements the Boltzmann distribution could bring to exploration compared to the unimodal Gaussian distribution that is commonly used in continuous action deep RL algorithms. The fact that StaQ seems better at solving these MDPs than the backbone algorithms used by Tang et al. might allow for a better investigation of these questions in the future.
>
> **Q2. Discretization on MuJoCo alters dynamics and conclusions do not directly transfer.** We only put a wrapper on MuJoCo's action space to translate action indices into action vectors. After that, the step function of the original MuJoCo env is used and so the dynamics are exactly the same. The more limited action space means that performance is expected to be lower than when using continuous action RL algorithms, but our new experiments in Appendix E show that using a finer grained discretization can further close the gap with said algorithms.
>
>
> *Reference:*
> * Tang et al.; Discretizing Continuous Action Space for On-Policy Optimization; AAAI, 2020

---

### Official Review · Reviewer_UZAB · 2025-11-01

**Soundness:** 3
**Presentation:** 2
**Contribution:** 4
**Rating:** 6
**Confidence:** 3

**Summary:**

Policy mirror descent (PMD) algorithms can average out policy-evaluation errors, but their exact implementation is impractical since it requires storing all past Q functions. To address this, the paper proposes a finite-memory variant, called StaQ, which only requires retaining the last $M$ Q functions. The paper shows that the policy update error vanishes for sufficiently large $M$ and the number of iterations, and the averaging effect in PMD can be obtained by introducing a KL divergence penalization during policy update.

**Strengths:**

The finite-memory formulation is well-motivated, enabling a practical implementation of the PMD algorithm without requiring infinite storage or recomputation.

The paper discusses the similarities and differences between the proposed method and related works, mainly in Section 2, and further connects relevant prior studies throughout later sections where appropriate.

The work is supported by theoretical results. Theoretical analysis for the convergence of Entropy-regularized Policy Mirror Descent (EPMD) is provided in Section 4, to connect EPMD to finite memory EPMD and improve over existing analysis on the convergence rate of EPMD. Theorem 5.1 extends the convergence analysis to the finite memory setting.

The paper considers the applicability of the algorithm by providing practical implementation for EPMD in Section 4.3, and for finite memory EPMD in Section 5.1 (StaQ).

The paper empirically investigates how the number of stored Q functions, $M$, affects StaQ’s performance, demonstrating when the finite-memory approach approximates the exact EPMD’s behavior.

The method is empirically evaluated across a wide range of tasks, and detailed parameter settings are reported to support reproducibility.

**Weaknesses:**

The paper could be strengthened by quantifying memory usage and explicitly analyzing the trade-off between memory consumption and learning efficiency. StaQ lies between natural policy gradient methods (e.g., TRPO) and PMD. While StaQ reduces storage and computation by keeping only $M$ recent Q functions compared to PMD, it may still require higher memory usage compared to other baselines like TRPO. It may be worth comparing the memory usage of these methods and their learning curves.

The runtime in Table 1 provides limited insight into StaQ’s advantages. It does not capture one of the key benefits of StaQ, which lies in maintaining stable and accurate policy updates under limited memory. Table 1 mainly suggests that varying the number of stored Q functions does not have a large effect on the running time. Including learning curves in the main paper, to highlight StaQ’s learning efficiency, might make the empirical case more convincing.

The method is limited to discrete action space tasks. For continuous control tasks, the paper discretizes the action space, which may lead to loss of precision and suboptimal policies.

**Small thing:**

Line 324-325: “...over states sampled from _some a_ predefined initial state distribution.” You may want to remove _some_.

**Questions:**

The choice of $M$ may depend on the environment. Do you expect the observed pattern, where StaQ with M=300 approximates PMD, to generalize to other environments, such as those with higher stochasticity or delayed rewards?

In Figure 2, Humanoid-v4, M=500 performs worse than M=300. Similarly, in Table 2, M=300 occasionally outperforms the exact PMD. This result seems to contradict the intuition that a larger M approximates the exact PMD better. Where do you think the StaQ’s performance advantage may come from?

---

> ### Author Response · Authors · 2025-11-21
> **Weaknesses**
>
> Dear reviewer, thank you for your thorough review and for all the kind words regarding our theoretical contributions, our practical implementation and our empirical study. We address below the weaknesses you highlighted and the questions you asked.
>
> **W1. Memory consumption.**
>
> Memory use grows linearly w.r.t. M. The largest model for the medium-sized environments we considered in this paper was around 10Mb (for MinAtar tasks) and hence for $M=300$, StaQ requires around 3Gb of GPU memory. You are indeed right that the GPU memory usage is larger than that of the baselines, but at least in the tasks we tested it on it fitted comfortably in any modern GPU. To add to that, we point out that during policy evaluation, the StaQ *need not* be in the GPU memory as we do forward and backward passes only on a single Q-function. The StaQ of Q-functions need to only be present in GPU memory during data gathering, where we only process a few states at a time (or a single one if environments are not parallelized) and where we do not need to store replay buffers or gradient information in GPU memory. As such, when the full StaQ is needed, almost all of the GPU memory can be dedicated to it, further reducing the risks of being out of memory. In the future, to scale to very large Q-networks, one could look into adding feature sharing capabilities such that not all layers are modified every iteration and can thus be shared, reducing memory and computation, at the expense of a more complex computational graph and truncation mechanisms.
>
> **W2. More experimental results in main paper. Plot performance vs memory.**
>
> Taking your comment into account, we have updated the paper to include experimental results in the main paper. Please see the global comment for a detailed list of changes. In particular, we have increased the number of seeds to 20 and added a plot of the aggregated final performance across tasks in Fig. 2 of the main paper. Fig. 2 in addition to Fig. 4 in Appendix B have $M$ in the y-axis and demonstrate the performance tradeoff vs $M$. While they do not include memory usage explicitly, since the memory overhead is linear w.r.t. $M$ (with the NatGrad baseline being equal to $M=2$ for the extra policy network), the performance payoff vs memory can be inferred straightforwardly.
>
>
> **W3. Suboptimal policies due to action space discretization.**
>
> The generalisation to the continuous-action setting would indeed be interesting, and we have added Appendix E in our updated manuscript discussing promising ways of extending StaQ to the continuous-action setting, via e.g. the Gumbel-max trick as proposed by Reviewer E3vT or by distilling the StaQ into an actor network.
>
> However, in general these approaches introduce an additional source of error compared to the discrete-action setting where we can sample directly from the policy. We therefore believe that the discrete-actions setting allows us to isolate the impact of the policy update step, independent of the specific choice of actor implementation and the additional optimization errors it introduces.
>
> The discretisation of continuous-action tasks can present interesting research direction in itself. We have added an additional experiment (see Fig. 10) where we discretize the action space based on Tang et al. 2020, as follows. We discretize each action dimension into a finite number of atoms and assume that each action dimension follows an independent distribution---which is also assumed by continuous action RL algorithms using diagonal Gaussian policies. In our setting, this translates to the logit space, which is also the space of Q-functions, by assuming separability of the Q-function approximation i.e. that $q_k$, the approximation of the Q-function $Q_k$, is given by $q_k(s,a) = \sum_{i=1}^{\text{dim(A)}} q_k^{[i]}(s, a^{[i]})$, where $a^{[i]}$ is entry $i$ of the action vector $a$ and $\{q_k^{[i]}\}_{i=1}^{\text{dim}(A)}$ is a set of real functions. This independence assumption makes it possible to consider fine grained discretizations with extremely large but finite action spaces, from which one can sample exactly and efficiently by sampling each action dimension independently.
>
> We can see large improvements due to the finer grained discretization on several environments. Moreover, using StaQ improves over previous results reported by Tang et al. 2020 that used PPO and TRPO as base learners on most Mujoco tasks, showing the impact our work could have in this line of research. While such discretization could still lead to a loss of precision as you say, it raises interesting research questions on the improvements the Boltzmann distribution could bring to exploration compared to the unimodal Gaussian distribution that is commonly used in continuous action deep RL algorithms. The fact that StaQ seems better at solving these MDPs than the backbone algorithms used by Tang et al. might allow for a better investigation of these questions in the future.

---

> > ### Author Response · Authors · 2025-11-21
> > **Questions**
> >
> > **Q1. Choice of M vs environments: higher stochasticity and delayed rewards.**
> > In general we expect that the number of Q-functions required to depend on the degree of policy evaluation error. Indeed, in some early experiments on Classic Control, we observed that we were able to approximate Exact PMD with $M$ in the tens instead of in the hundreds. This still shows to some extent on MuJoCo tasks that exhibit less differences between $M=100$ and Exact PMD, as opposed to the higher dimensional MinAtar tasks where larger $M$s are needed. Therefore, it appears that the easier it is to approximate the Q-function the more $M$ can be reduced. By extension, if learning the Q-function is made more complex by the presence of excessive noise or of delayed rewards than we also expect higher $M$ to be beneficial.
> >
> > **Q2. Variance in performance for $M\ge300$.**
> > In the updated version of the manuscript, we have added aggregated performance plots (now Fig. 2), which aggregate over tasks (95% stratified bootstrap CIs, following the recommended evaluation protocol of Agarwal et al, 2021).
> >
> > Based on these new results and Fig. 4 that replaces the previous Table 2 and better shows the noise in final policy performance, we believe that the differences between different values of $M\ge 300$ are primarily due to the inherent stochasticity of Deep RL. For example, in the Humanoid-v4 you mentioned, in the new Table 2, the results are $5143 ± 90$ for $M=300$ and $5102 ± 132$ for $M=500$: there is a significant overlap in the confidence intervals.
> >
> > While it is indeed possible that there is an advantage in e.g. $M=300$ over $M=500$, due to the faster deleting of early Q-functions, we believe that in our experiments, any remaining difference after accounting for statistical variation is very small compared to the difference in performance between e.g. $M\le100$ and $M\ge 300$. Given the current empirical evidence, we would thus not recommend optimizing for $M$ but simply picking the highest $M$ given the environment/GPU, saving precious hyper-parameter tuning time.
> >
> >
> > *References:*
> > * Agarwal et al.; Deep Reinforcement Learning at the Edge of the Statistical Precipice; NeurIPS, 2021
> > * Tang et al.; Discretizing Continuous Action Space for On-Policy Optimization; AAAI, 2020

---

### Official Review · Reviewer_E3vT · 2025-11-01

**Soundness:** 3
**Presentation:** 3
**Contribution:** 2
**Rating:** 6
**Confidence:** 4

**Summary:**

The paper studies policy mirror descent (PMD) with a finite memory of past Q-functions in discrete action spaces. It proposes StaQ, an optimization-free policy update rule implemented through a stacked neural network.Policy mirror descent algorithms usually include a distance term, such as KL divergence, to enforce a trust region and ensure stable updates.

Theoretically, the paper shows that if there is no policy evaluation error and the memory size is large enough, the finite-memory update converges to the same policy as exact policy mirror descent. It also retains the error-averaging effect, with extra terms that decay exponentially with memory size.

The analysis covers both value iteration and policy iteration, providing explicit bounds that separate the effects of evaluation error and truncation. Empirically, the results show that increasing memory size improves performance up to a plateau, matching the exact-PMD baseline within the training horizon on discretized MuJoCo and MinAtar benchmarks.

**Strengths:**

- Clear problem formulation and principled update:
  - The finite-memory PMD policy is
    $$
    \xi\_{k+1} = \beta\\xi\_k + \alpha q\_k + \frac{\alpha \beta^M}{1-\beta^M}\big(q\_k - q\_{k-M}\big),
    $$
    is derived cleanly and highlights both the deletion of the oldest $q$ and the mild overweighting of the latest $q$.
- Theoretical extensions to Vieillard et al. (2020a) and careful bounds:
  - The paper correctly qualifies that if policy evaluation error vanishes and $M$ satisfies the stated condition, the update introduces no additional policy-update error, i.e., the algorithm converges to the optimal entropy-regularized policy.
- Practical relevance and engineering:
  - The stacked-neural-network (SNN) implementation exploits batched evaluation of many frozen $Q$-networks on GPU, making large $M$ feasible in practice. Precomputing logits for the replay buffer further amortizes cost during policy evaluation.
  - Wall-clock data (Table 1) show minimal runtime increase with $M$ on mid-scale tasks, supporting the feasibility claim.
- Empirical evidence aligns with their theorems:
  - Across discrete-action MuJoCo and MinAtar, increasing $M$ generally improves performance until diminishing returns in some environments; large $M$ becomes empirically close to the exact-PMD variant (that never deletes within the 5M-step window).

**References**
- Nino Vieillard, Tadashi Kozuno, Bruno Scherrer, Olivier Pietquin, Remi Munos, and Matthieu Geist. Leverage the average: an analysis of kl regularization in reinforcement learning. In Advances in Neural Information Processing Systems, 2020a.

**Weaknesses:**

- The main results and implementation are limited to discrete action spaces. While the paper acknowledges this, a discussion of algorithmic options for continuous actions (e.g., approximate sampling or parameterized policies) would strengthen the practical impact.
- Empirical analysis gaps:
  - Improvement with $M$ is not monotone across all tasks. The paper frames the trend as "beneficial effects with diminishing returns”, but the figures/tables show non-monotonic or even worse behavior on some environments (e.g., Hopper-v4, Breakout-v1, Freeway-v1, HalfCheetah-v4). The analysis could more explicitly discuss when/why performance can dip at intermediate $M$.
  - Baselines: Mirror Descent Policy Optimization (MDPO) is a closely related comparator for policy mirror descent with approximate updates; including it would position StaQ more clearly against the most directly related policy-optimization methods.
  - Sensitivity and interpretability: Although $\epsilon$-softmax exploration is used uniformly in the main study and a “sticky” behavior policy is shown to unlock MountainCar in a focused analysis, a broader sensitivity study of behavior policy design and its interaction with error averaging would be valuable.
- Benchmark choice: Discretized MuJoCo is not ideal; full Atari or other naturally discrete benchmarks would be more appropriate.

**Questions:**

1. The paper’s methods and theory are limited to discrete action spaces. How difficult would it be to extend StaQ to continuous actions? Could re-parameterized actions such as Gumbel softmax policies preserve similar theoretical guarantees?
2. The paper assumes vanishing policy evaluation error for convergence proofs. How sensitive is StaQ to nonzero evaluation errors in practice, and could the analysis be extended to handle this more realistic case?
3. The improvement with memory size (M) is not monotonic across tasks. Can the authors explain why intermediate (M) sometimes leads to worse performance? Does truncation interact with learning dynamics or noise in Q-function estimates?
4. The experiments lack baselines such as Mirror Descent Policy Optimization (MDPO). Why was MDPO omitted, and how might StaQ compare to it theoretically and empirically?
5. The paper uses only one behavior policy variant (ε-softmax) except for a small analysis on sticky policies. Could more diverse behavior policies change StaQ’s error-averaging behavior or its convergence?
6. Given the strong theoretical results but limited empirical exploration, how confident should we be that StaQ’s convergence properties translate to real-world deep RL settings?
7. Are there scenarios where finite memory could harm performance due to outdated Q-functions being overweighted or reintroduced?

---

> ### Author Response · Authors · 2025-11-21
> **Continuous action spaces**
>
> Dear reviewer, we thank you for your in depth assessment of our work and detailed feedback. Below we provide a detailed answer to the question regarding continuous action spaces in a first post as that was asked about other reviewers too. In a followup post we answer the remainder of your questions and detail the new experiments and modifications made to the original submission.
>
> **1. Continuous actions and Gumbel-max trick?**  Thank you for the suggestion. Using the Gumbel-max trick could indeed be a way to sample from StaQ's policy in the continuous action case. For the theory to hold, we only need accurate samples when computing the targets for the current Q-function (and these sampled actions can be pre-computed), meaning that during data gathering, the behavior policy could use less accurate samples from StaQ's policy to speed-up inference. Nonetheless, we expect this solution as well as the alternative of distilling the StaQ into a more compact actor to yield additional policy update errors compared to exact PMD and StaQ. As discussed in the paper (line 111 and Sec. 4.3), Zhan et al. 2023 showed that PMD with approximate policy updates have a higher error floor (see Thm. 2 in Zhan's paper and the dependence to $\epsilon_\text{opt}$) and might converge to lower quality solutions. As such, these extensions do not fit the message of the paper, which is that for some environments and thanks to the continuous advances in GPU hardware, it can become more advantageous to stack Q-functions to get an error free policy update with low computational overhead, instead of following the more traditional actor-critic approach to PMD.
>
> To tackle continuous actions following the conclusions of our paper, one could use StaQ instead as the gating policy in a hierarchical RL framework. For instance, starting from our Mujoco experiments, one could consider the hand discretized actions we used as sub-policies and further optimize them via gradient ascent. The sub-policies could be more complex, for instance PID controllers or neural based as in the work of Gehring et al. 2021, or Hafner et al. 2022, that use a discrete set of sub-policies and would thus preserve the above properties of fast and accurate policy updates for the gating policy if optimized by truncated PMD.
>
> Yet another direction is to leverage an independence assumption in the policy following Tang et al. 2020. Concretely, Tang et al. discretize each action dimension into a finite number of atoms and assume that each action dimension follows an independent distribution---which is also assumed by continuous action RL algorithms using diagonal Gaussian policies. In our setting, this translates to the logit space, which is also the space of Q-functions, by assuming separability of the Q-function approximation i.e. that $q_k$, the approximation of the Q-function $Q_k$, is given by $q_k(s,a) = \sum_{i=1}^{\text{dim(A)}} q_k^{[i]}(s, a^{[i]})$, where $a^{[i]}$ is entry $i$ of the action vector $a$ and $\{q_k^{[i]}\}_{i=1}^{\text{dim}(A)}$ is a set of real functions. This independence assumption makes it possible to consider fine grained discretizations with extremely large but finite action spaces, from which one can sample exactly and efficiently by sampling each action dimension independently. Incidentally, this separability assumption could allow a more efficient implementation of the Gumbel-max trick you suggested since we can maximize each dimension separately.
>
> To illustrate how this could benefit StaQ, we have implemented the discretization from Tang et al. 2020 in combination with StaQ, and we show the results in Appendix E. We can see large improvements due to the finer grained discretization on several environments. Moreover, using StaQ improves over previous results reported by Tang et al. 2020 that used PPO and TRPO as base learners on most Mujoco tasks, showing the impact our work could have in this line of research.
>
> In summary, beyond environments that have naturally discrete action spaces or low dimensional continuous action spaces that can be easily discretized, we have outlined several RL frameworks that tackle continuous actions spaces yet leverage Boltzmann policies and that could benefit from our work. We think interesting research questions in this direction is to investigate the improvements the Boltzmann distribution could bring to exploration compared to the unimodal Gaussian distribution that is commonly used in continuous action deep RL algorithms.
>
> We thank you for your question and we have added this discussion in the paper.
>
> *References:*
> * Zhan et al.; Policy mirror descent for regularized reinforcement learning: A generalized framework with linear convergence; SIAM Journal on Optimization, 2023
> * Gehring et al.; Hierarchical skills for efficient exploration; NeurIPS, 2021
> * Hafner et al.; Deep Hierarchical Planning from Pixels; NeurIPS, 2022
> * Tang et al.; Discretizing Continuous Action Space for On-Policy Optimization; AAAI, 2020

---

> ### Author Response · Authors · 2025-11-21
> **Questions 2-6**
>
> **2. Vanishing evaluation error assumption.** We never made such an assumption. As we say in line 253, the error term that depends on evaluation errors, $\sum_{i=0}^{k}\gamma^i ||{E_{k-i}}||$, may not vanish. The advantage of PMD is that $E_{k-i}$ itself is a weighted average of the per iteration errors $\{\epsilon_i\}_{i=0}^k$ which might reduce the magnitude of the error if the per iteration errors cancel each other. This property was already discussed in Viellard et al. 2020, and we have extended it to a different Bellman operator, to the policy iteration case and to the case of finite $M$, where we showed that the averaging of error property still holds up to some extra terms that vanish exponentially fast w.r.t. $M$.
>
> **3. Non-monotonic improvements w.r.t. $M$.** We have substantially improved our main experiment by increasing the number of seeds and using aggregate metrics to reduce the inherent noise of deep RL. Please see the global comment for a detailed list of changes. We believe now the almost monotonic dependency w.r.t. $M$ is more clear, especially on MinAtar tasks, as can be seen in Fig. 2, and Fig. 4 that additionally shows the magnitude of said noise. While it might appear that, for example on Humanoid-v4, $M=300$ and $M=500$ have better end performance than exact PMD in Fig. 4 using 20 seeds, Table 2 shows that this conclusion does not hold anymore with 30 seeds.
>
> From a theoretical point of view, nothing stops intermediary $M$ values to have lower error floors than exact PMD. For instance, the truncation from a finite $M$ could eliminate earlier Q-functions that might have larger evaluation errors as suggested by your Question 7 and Question 2 of Reviewer f4Z1. However, from our experiments, we did not notice such an effect, and even if improvements of truncated PMD over full sum PMD that can be seen on some tasks turns out to be true, it is still extremely minor. With the current empirical evidence, we do not recommend tuning for $M$ but simply picking as large an $M$ as the task/GPU allows, as we think the potential improvement to performance are not worth the extra computational cost of hyper-parameter tuning.
>
> **4. MDPO?** MDPO uses adaptive KL weight which is different from our setting. We would have happily added it as a baselineat least for the broader comparison to deep RL in Appendix B.3, however the existing off-policy implementation is built on top of SAC, and only supports continuous action environments.
>
> We are currently investigating whether we can implement an actor-critic baseline inspired by MDPO within our code base for discrete actions. We will report back if we have new results.
>
> **5. Impact of behavior policy.** Exploration of course impacts online deep RL algorithms. While we would want to design an experiment where $M$ only impacts the Bellman operator when computing the Q-function targets, which is what our analysis is about, in practice different $M$ values also change the behavior policy. We have considered performing experiments in an offline RL setting, to separate the impact of $M$ on the behavior policy vs the Bellman operator, however, offline RL algorithms generally need extra terms to minimize deviation from dataset actions which are outside the scope of this paper. We will look at further experiments regarding this question and will provide an update if we have new results. In the current state, we still have used relatively standard exploration heuristics, sampling from Boltzmann distributions and adding $\epsilon$-greedy exploration which further reduces differences between behavior policies: indeed from the convexity of the KL between distribution pairs, we have that for two policies $\pi$ and $\pi'$, their mixture with a uniform brings them closer together, i.e. $\text{KL}\left((1-\epsilon)\pi + \epsilon \text{ Unif.}(A), (1-\epsilon)\pi' + \epsilon \text{ Unif.}(A)\right) \leq (1-\epsilon) \text{KL}(\pi, \pi')$. In our internal testing, adding $\epsilon$-greedy also had a generally positive impact in performance on some tasks compared to using $\epsilon=0$.
>
> **6. Strong theory but limited empirical exploration.** We thank you for the kind words regarding the theoretical contributions. Regarding the experiments, we have made large revisions to the experiments to improve their breadth and their statistical significance. We will continue to update our manuscript throughout the rebuttal to incorporate reviewers' feedback, and we hope you can let us know whether we have improved over this point by the end of the rebuttal period.

---

> > ### Author Response · Authors · 2025-11-21
> > **Question 7**
> >
> > **7. Harmed performance due to outdated Q-functions and overweight.** We hope to understand correctly what you mean by outdated Q-functions and overweight. If by overweight you mean the weight correction $\frac{1}{1-\beta^M}$ due to the truncated sum that appears in our policy (Eq. 2) and is not in vanilla PMD (Eq. 1), there are two ways in which outdated Q-functions and the overweight do not negatively impact each other: if $M$ is small, then the overweight is high but the sum truncates very old Q-functions. If $M$ is larger, then the sum might include old Q-functions but the overweight is small and depending on $\beta$ its contribution to the policy might be heavily discounted. Reviewer f4Z1 asked about the impact of initial Q-functions that are of low quality. We agree that initial Q-functions due to the limited amount of gradient updates they receive might have higher evaluation errors. However, in StaQ and in most deep RL implementations, Q networks are initialized to very flat low magnitude functions over the state-action space. As such, since they are flat functions they do not influence the policy one way or another, since the policy is invariant to a constant shift to its logits. If initial Q-functions had a bigger impact on the algorithm, then we believe a finite $M$ would more clearly outperform the untruncated PMD, but this does not show in a significant way in our experiment.

---

> > > ### Author Response · Authors · 2025-12-02
> > > **Follow-up experiments regarding Question 4 and 5**
> > >
> > > Dear reviewer/AC, we follow-up on our previous response to include new experiments you requested in Question 4 and 5, and hope we have as a result satisfied all of your requests.
> > >
> > > In regards to **Question 4**, you asked for more baselines and specifically for MDPO. In approximate entropy regularized PMD, the common approach to cope with the infinite sum over past Q-functions is to use an actor-critic approach with an explicit policy network. We have added a new set of experiments in Appendix C.2 comparing our algorithm to three additional Actor-Critic (AC) PMD baselines (complementing our prior comparisons to the AC natural gradient baselines):
> > > 1. The first AC baseline is similar to SAC (**Haarnoja et al.**; **Soft Actor-Critic: Off-Policy Maximum Entropy Deep Reinforcement Learning with a Stochastic Actor**; ICML 2018). As discussed in what is now Appendix C.1, StaQ with $M=1$ can be seen as an adaptation of SAC to discrete action spaces as it has entropy regularization but no KL regularized policy update. In discrete action spaces, one can sample directly from the policy proportional to $\exp\big(\frac{q_k}{\tau}\big)$, as done by StaQ with $M=1$, however, for completeness we also implemented an AC method that further minimizes the same KL loss as SAC, i.e.   $\pi_{k+1} = \underset{\pi}{\arg\min}\ \mathbb{E}_{s\sim {\cal D}}\left[\text{KL}\left(\pi(s) \middle| \frac{\exp\left(\frac{q_k(s)}{\tau}\right)}{Z}\right)\right]$, where $\cal D$ is the replay buffer.
> > > 2. The second AC PMD baseline follows the policy update of ECPO (**Mei et al.**; **On Principled Entropy Exploration in Policy Optimization**; IJCAI 2019), which minimizes the M-Projection between the closed form solution of entropy regularized PMD and the actor, resulting in the update
> > > $\pi_{k+1} = \underset{\pi}{\arg\min}\ \mathbb{E}_{s\sim {\cal D}}\left[\text{KL}\left(\frac{\pi_k^\beta \exp\left({\alpha q_k}\right)}{Z} \middle| \pi(s) \right)\right]$. The authors argue that minimizing the M-Projection preserves the support of the distribution and prevents the premature elimination of actions during exploration.
> > > 3. The third AC PMD method follows the policy update of MDPO (**Tomar et al.**; **Mirror descent policy optimization**; ICLR 2022). Unlike ECPO, MDPO does not use the closed form solution of the PMD update, but directly optimizes via gradient ascent a version of the PMD objective (Eq. 9 in our paper) averaged over replay buffer states  $\pi_{k+1} = \underset{\pi}{\arg\max}\ \mathbb{E}_{s\sim {\cal D}} \left[q_k(s) \cdot \pi(s) - \tau h(\pi(s)) -\eta \text{KL}(\pi(s);\pi_k(s))\right]$. We said in our previous response that MDPO uses an adaptive KL regularization weight, and while the algorithm is presented this way, the implementation and experiments are done with a fixed KL weight and thus direct comparisons with StaQ are valid.
> > >
> > > In Appendix C.2, we discuss these baselines in more detail, include a sensitivity analysis of key hyperparameter, and a 20 seed comparison with StaQ on MinAtar. Results show that the KL regularized AC methods (i.e. those following ECPO and MDPO's update) have performance between that of StaQ with $M=1$ and $M=50$, while AC-NoKL performs worse than $M=1$. Moreover, because of the extra actor training, these AC methods end up being slower to even StaQ with $M=300$, as shown by the table below, while the setting $M=300$ was found to perform best on this domain.
> > >
> > > |                          | M=1 | M=100 | M=300 | AC-NoKL | AC-MProj | AC-DirectOpt |
> > > |--------------------------|-----|-------|-------|---------|----------|--------------|
> > > | MinAtar/SpaceInvaders-v1 | 9.6 | 10.4  | 10.5  | 13.9    | 13.1     | 13.2         |
> > >
> > > We believe these results further reinforce our claims that thanks to efficient GPU based implementations using batched operations, it can be for discrete action spaces and on some medium-sized environments, more advantageous to "stack" Q-functions as we propose in this paper instead of using an AC method, both for the sound theoretical properties, the empirical performance and the shorter run time.
> > >
> > >
> > > Regarding **Question 5**, we have added a new set of experiments in Appendix D.1, comparing the impact of different exploration noise $\epsilon$ as you requested. Results show that on some environments, $\epsilon$ exploration is not necessary and might even harm performance if set too high, but helps on others. Results also show that our choice of hyper-parameter value appears sound, and for all choices of $\epsilon$ we are able to match exact PMD performance with a finite $M$ as in our main experiment.

---

### Author Response · Authors · 2025-11-12
**Updated experiments**

We thank the reviewers for their thorough feedback on both the theoretical and experimental aspects of our submission. We are also grateful for the unanimous positive assessment of our theoretical contributions.

Several reviewers noted concerns regarding the placement of our experiments in the appendix (UZAB and f4Z1) and mentioned that the monotonic improvement with respect to M was not immediately clear from the figures and tables (E3vT). Using the additional page and time provided since the initial submission, we have substantially revised our main experiment to address these comments. Below is a summary of the changes:

* **Increased number of seeds:** we raised the number of random seeds from 5 to 20 for all baselines and all values of M, thereby reducing noise in the final policy performance.

* **Aggregated plots:** following the recommendations of Agarwal *et al.* (*Deep Reinforcement Learning at the Edge of the Statistical Precipice*, NeurIPS 2021), we now include aggregated plots to further reduce noise and facilitate interpretation of the results.

* **Main paper self-containment:** these aggregated plots are now included in the main paper, making it more self-contained and easier to interpret without reference to the appendix.

* **Appendix updates:** in Appendix B.1, learning curves now display 95% confidence intervals, which are tighter due to the increased number of seeds, improving readability. We have also replaced the table of final policy performances with per-task box plots that better capture data uncertainty and allow for more granular analysis in relation to the aggregated plots of the main paper.

* **New baseline:** We added a new baseline that integrates TRPO’s line search into the previous NatGrad baseline (details provided in blue text in the main paper). This ensures that all algorithms in the main experiment use identical policy evaluation procedures, isolating the effects of different policy update mechanisms—the main focus of this work.

We believe these revisions address the reviewers’ above concerns and more clearly support our observation that increasing M leads to an almost monotonic improvement in final policy performance, with diminishing returns. We note, however, that even with 20 seeds, some noise remains. For example, on *Humanoid-v4*, Exact PMD appears to have slightly worse mean performance in Fig. 3 (and median in Fig. 4), but this discrepancy disappears in Table 2, where the same experiment was repeated with 30 seeds.

---

### Author Response · Authors · 2025-12-03
**Rebuttal Summary**

Dear AC, Dear Reviewers,

We thank all reviewers for their thoughtful and constructive feedback. The discussion has substantially strengthened the paper, and we are grateful for the uniformly positive assessment of our theoretical contributions. Below we summarise the main concerns and how the revised manuscript addresses them.
1. **Experimental results in the main paper & monotonicity w.r.t. M.**
Reviewers E3vT, UZAB, and f4Z1 raised concerns regarding the lack of main-paper experimental results in our initial submission, and/or unclear performance trend with respect to M. In the revision, we significantly expanded and clarified the empirical section: the **main paper** now includes aggregated performance curves (**Fig. 2**), over 20 random seeds to improve statistical confidence. These results show a clearer performance improvement as M increases, with diminishing returns as StaQ approaches the behaviour of exact Policy Mirror Descent—as suggested by Thm. 5.1.  We believe the new experimental section now provides a compelling and comprehensive empirical validation of our theoretical insights.
2. **MDPO and additional AC baselines.**
Reviewer E3vT asked about comparisons with MDPO. The original MDPO implementation relies on a SAC-based off-policy continuous-action setup that is not directly applicable to our discrete-action setting. Nonetheless, to address the reviewer’s request, we have added new Actor–Critic baselines implementing the MDPO-style policy update, as well as ECPO and SAC variants. These baselines perform between StaQ with M=1 and M=50, and—as expected—incur higher runtime due to additional actor gradient steps. Full details are now provided in **Appendix C.2**.
3. **Discrete-action limitation.**
All reviewers noted that our current implementation of StaQ is restricted to discrete actions. Lifting this limitation is nontrivial because StaQ’s benefits fundamentally rely on sampling efficiently from the stacked Q-functions. To address this point, we added a detailed discussion in **Appendix E** outlining several promising directions for extending StaQ to continuous actions.
 Additionally, we now include new experiments (**Fig. 16**) using a fine-grained discretization approach inspired by Tang et al. (2020), which leads to markedly improved performance compared to the coarse discretization initially used. These results also outperform prior PPO/TRPO baselines reported by Tang et al., suggesting that StaQ could provide a promising foundation for future advances in this line of work.

---
In summary, we believe the paper now offers substantially improved and well-supported empirical evidence to support our strong theoretical contributions, and clarifies connections to prior Actor–Critic and PMD-related works. Given the strengthened experiments, clearer positioning, and the practical relevance of StaQ under modern computational resources, we hope the reviewers agree that the revised paper makes a solid and timely contribution.

---

### Meta-Review · Area_Chair_yJwX · 2026-01-05

**Summary:**

In this paper, the authors propose StaQ, an optimization-free policy update rule implemented through a stacked neural network. StaQ becomes the base for their discrete action policy mirror descent (PMD) algorithm, which uses a finite memory of past M (a hyper-parameter) Q-functions. This is in contrast to PMD algorithms that usually include a KL-term to enforce a trust-region, and thus, ensure stable updates. They theoretically show that with no policy evaluation error and large enough memory size, the finite-memory update converges to the same policy as exact PMD. Finally, they empirically (using discretized MuJoCo and MinAtar benchmarks) show that increasing memory size improves performance up to matching the exact PMD.

I think the paper does a good job in proposing an optimization-free policy update rule, showing how it can be approximately implemented using a finite memory of past M Q-functions, and theoretically and empirically show that as M becomes large enough, the proposed method converges to the exact PMD solution (under no policy evaluation error). However, the paper does not properly explain/highlight the limitations of the method, most importantly being restricted to discrete actions. Moreover, it neither properly discusses the relationship between the proposed approach nor algorithmically and empirically compares it with the most related algorithm MDPO. MDPO is a framework of practical on-policy and off-policy PMD algorithms, which has been applied to both MuJoCo and Atari benchmarks. I believe a thorough discussion on the relationship between these algorithms and a comprehensive empirical comparison between them are needed and can significantly improve the quality of the paper. I do not think what was added to the appendix in response to the reviewers' comments on MDPO during the rebuttals is sufficient. Finally, I would suggest that the authors take other reviewers' comments regarding the experiments into account, improve their work, and prepare it for future venues. I believe the work has merit to be published but requires more work.

**Reviewer Concerns:**

The authors addressed some of the reviewers' concerns: 1) changing the structure of the paper to include more experimental results within its main body, 2) aggregating over more random seeds to improve statistical confidence, 3) adding a discussion on several promising directions for extending StaQ to continuous actions, and 4) including some discussions on the relation with MDPO.

However, I do not think their main concerns, namely 1) being limited to discrete actions, 2) lack of a proper comparison (algorithmically and empirically) with MDPO, as practical policy mirror descent algorithms for on-policy and off-policy learning, and 3) the choice of benchmarks (discretized MuJoCo and simple MinAtar games instead of a large number of Atari games used in other papers), were fully addressed during the rebuttals.

**Reviewer Scores:**

Three of the reviewers did not participate in discussion with the authors. Reviewer f4Z1 interacted with them but did not find their responses convincing enough to raise their score.

---

### Decision · Program_Chairs · 2026-01-26

Reject